# Improving Flow Matching by Aligning Flow Divergence

**Yuhao Huang** [1 2]  **Taos Transue** [1 2]  **Shih-Hsin Wang** [1 2]  **William Feldman** [1]  **Hong Zhang** [3]  **Bao Wang** [1 2]

## Abstract

Conditional flow matching (CFM) stands out as an efficient, simulation-free approach for training flow-based generative models, achieving remarkable performance for data generation. However, CFM is insufficient to ensure accuracy in learning probability paths. In this paper, we introduce a new partial differential equation characterization for the error between the learned and exact probability paths, along with its solution. We show that the total variation gap between the two probability paths is bounded above by a combination of the CFM loss and an associated divergence loss. This theoretical insight leads to the design of a new objective function that simultaneously matches the flow and its divergence. Our new approach improves the performance of the flow-based generative model by a noticeable margin without sacrificing generation efficiency. We showcase the advantages of this enhanced training approach over CFM on several important benchmark tasks, including generative modeling for dynamical systems, DNA sequences, and videos. Code is available at Utah-Math-Data-Science.

## 1. Introduction

Flow matching (FM) – leveraging a neural network to learn a predefined vector field mapping between noise and data samples – has emerged as an efficient simulation-free training approach for flow-based generative models (FGMs), achieving remarkable stability, computational efficiency, and flexibility for generative modeling (Lipman et al., 2023; Albergo & Vanden-Eijnden, 2023; Liu et al., 2023). Compared to the classical likelihood-based approaches for training FGMs, e.g., (Chen et al., 2018; Grathwohl et al., 2018),

[1]Department of Mathematics, University of Utah, Salt Lake City, UT, USA [2]Scientific Computing and Imaging (SCI) Institute, Salt Lake City, UT, USA [3]Mathematics and Computer Science Division, 240 Argonne National Laboratory, Lemont, IL, USA. Correspondence to: Bao Wang <wangbaonj@gmail.com>.

*Proceedings of the 42$^{nd}$ International Conference on Machine Learning*, Vancouver, Canada. PMLR 267, 2025. Copyright 2025 by the author(s).

FM circumvents computationally expensive sample simulations to estimate gradients or densities. The celebrated diffusion models (DMs) with variance preserving (VP) or variance exploding (VE) stochastic differential equations (SDEs) (Song et al., 2020) can be viewed as special cases of FGMs with diffusion paths (c.f. Section 2). Furthermore, FM excels in generative modeling on non-Euclidean spaces, broadening its scientific applications (Baker et al., 2024; Chen & Lipman, 2024; Jing et al., 2023; Bose et al., 2024; Yim et al., 2024; Stark et al., 2024).

At the core of FM is the idea of regressing a vector field that interpolates between the prior noise distribution $q(\boldsymbol{x})$ – typically the standard Gaussian – and the data distribution $p(\boldsymbol{x})$. Specifically, we aim to regress the vector field $\boldsymbol{u}_t(\boldsymbol{x})$ that guides the probability flow $p_t(\boldsymbol{x})$ interpolating between an easy-to-sample noise distribution and the data distribution, i.e., $p_0 = q$ and $p_1 \approx p$. The relationship between $\boldsymbol{u}_t$ and $p_t$ is formalized by the following continuity equation (Villani et al., 2009):

$$\frac{\partial p_t(\boldsymbol{x})}{\partial t} + \nabla \cdot (p_t(\boldsymbol{x})\boldsymbol{u}_t(\boldsymbol{x})) = 0.$$

FM approximates $\boldsymbol{u}_t$ using a neural network-parameterized vector field $\boldsymbol{v}_t(\boldsymbol{x}, \theta)$, seeking to minimize the FM loss:

$$\mathcal{L}_{\text{FM}}(\theta) := \mathbb{E}_{t, p_t(\boldsymbol{x})}\left[\left\|\boldsymbol{v}_t(\boldsymbol{x}, \theta) - \boldsymbol{u}_t(\boldsymbol{x})\right\|^2\right], \quad (1)$$

where $t \sim U[0, 1]$ follows a uniform distribution over the unit time interval $[0, 1]$.

However, equation (1) is intractable as $\boldsymbol{u}_t(\boldsymbol{x})$ is unavailable. To address this, an alternative simulation-free method, known as conditional flow matching (CFM) (Lipman et al., 2023; Albergo & Vanden-Eijnden, 2023), is employed. In CFM, $\boldsymbol{v}_t(\boldsymbol{x}, \theta)$ is trained by regressing against a predefined conditional vector field on a per-sample basis, ensuring both computational efficiency and accuracy. Concretely, for any data sample $\boldsymbol{x}_1 \sim p(\boldsymbol{x})$, we can define a conditional probability path $p_t(\boldsymbol{x}|\boldsymbol{x}_1)$ for $t \in [0, 1]$ satisfying $p_0(\boldsymbol{x}|\boldsymbol{x}_1) = q(\boldsymbol{x})$ and $p_1(\boldsymbol{x}|\boldsymbol{x}_1) \approx \delta(\boldsymbol{x} - \boldsymbol{x}_1)$, and define the associated conditional vector field $\boldsymbol{u}_t(\boldsymbol{x}|\boldsymbol{x}_1)$; see Section 2 for a review on several common designs of conditional probability paths. Once the conditional probability paths are defined, the marginal probability path $p_t(\boldsymbol{x})$ is given by:

$$p_t(\boldsymbol{x}) := \int p_t(\boldsymbol{x}|\boldsymbol{x}_1)p(\boldsymbol{x}_1)d\boldsymbol{x}_1.$$

Similarly, the marginal vector field is defined as:

$$\boldsymbol{u}_t(\boldsymbol{x}) := \int \boldsymbol{u}_t(\boldsymbol{x}|\boldsymbol{x}_1)\frac{p_t(\boldsymbol{x}|\boldsymbol{x}_1)p(\boldsymbol{x}_1)}{p_t(\boldsymbol{x})}d\boldsymbol{x}_1.$$

With these relations in mind, CFM regresses $\boldsymbol{v}_t(\boldsymbol{x}, \theta)$ against $\boldsymbol{u}_t(\boldsymbol{x}|\boldsymbol{x}_1)$ by minimizing the following CFM loss:

$$\mathcal{L}_{\text{CFM}}(\theta) := \mathbb{E}_{t,p(\boldsymbol{x}_1),p_t(\boldsymbol{x}|\boldsymbol{x}_1)}\left[\left\|\boldsymbol{v}_t(\boldsymbol{x}, \theta) - \boldsymbol{u}_t(\boldsymbol{x}|\boldsymbol{x}_1)\right\|^2\right]. \tag{2}$$

It has been shown that the CFM loss is identical to the FM loss up to a constant that is independent of $\theta$ (c.f. (Lipman et al., 2023)[Theorem 2]). Therefore, minimizing $\mathcal{L}_{\text{CFM}}(\theta)$ enables $\boldsymbol{v}_t(\boldsymbol{x}, \theta)$ to be an unbiased estimate for the marginal vector field $\boldsymbol{u}_t(\boldsymbol{x})$.

While CFM enables $\boldsymbol{v}_t(\boldsymbol{x}, \theta)$ to efficiently approximate $\boldsymbol{u}_t(\boldsymbol{x})$, we observe that their divergence gap[1] $|\nabla \cdot \boldsymbol{v}_t(\boldsymbol{x}, \theta) - \nabla \cdot \boldsymbol{u}_t(\boldsymbol{x})|$ can be substantial, resulting in significant errors in learning the probability path and estimating sample likelihood. Figure 1 highlights the challenges in learning a Gaussian mixture distribution using CFM. The full experimental setup for this result is provided in Section 3. Additionally, we will prove the existence of an intrinsic bottleneck of FM. This underscores the importance of improving FM for generative modeling, especially for tasks requiring accurate sample likelihood estimation. Indeed, such tasks are ubiquitous in climate modeling (Finzi et al., 2023; Wan et al., 2023; Li et al., 2024), molecular dynamics simulation (Petersen et al.), cyber-physical systems (Delecki et al., 2024), and beyond (Hua et al., 2024).

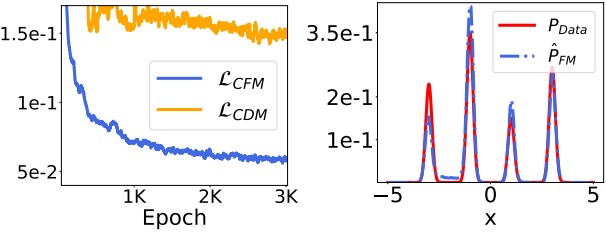

*Figure 1.* Experiments of training an FM model using CFM for sampling the 1D Gaussian mixture distribution in equation (18). The left panel shows that the conditional divergence loss $\mathcal{L}_{\text{CDM}}$ in equation (14) is much larger than the CFM loss $\mathcal{L}_{\text{CFM}}$, and the right panel shows the significant gap between the exact distribution ($p_{\text{Data}}$) and the distribution learned through FM ($\hat{p}_{\text{FM}}$).

### 1.1. Our Contribution

We summarize our key contributions as follows:

- We characterize the error between the exact ($p_t(\boldsymbol{x})$) and learned ($\hat{p}_t(\boldsymbol{x})$) probability paths using a partial differential equation (PDE); see Proposition 3.1. This new error

---

[1] Here, we use the absolute value notation since the divergence of the vector field is a scalar.

characterization describes how the error propagates over time, allowing us to derive a total variation (TV)-based error bound between the two probability paths; see Corollary 3.2 and Theorem 3.3. These theoretical results underscore the importance of controlling the divergence gap to enhance the accuracy in learning $\hat{p}_t(\boldsymbol{x})$.

- Informed by our established TV error bound, we develop a new training objective by combining the CFM loss with the divergence gap. However, directly minimizing the divergence gap is intractable since the divergence of the marginal vector field is unavailable. To address this issue, we propose a conditional divergence gap – an upper bound for the unconditional divergence gap. We refer to this new training objective as flow and divergence matching (FDM); see Section 4 for details.

- We validate the performance of FDM across several benchmark tasks, including synthetic density estimation, trajectory sampling for dynamical systems, video generation, and DNA sequence generation. Our numerical results, presented in Section 5, show that our proposed FDM can improve likelihood estimation and enhance sample generation by a remarkable margin over CFM.

### 1.2. Some Additional Related Works

The Kullback-Leibler (KL) divergence between the exact and learned distributions has been studied for DMs (c.f. (Song et al., 2021; Lu et al., 2022; Lai et al., 2023)) and FM (c.f. (Albergo et al., 2023)) with ODE flows, where it was observed that the FM loss in equation (1) alone is insufficient for minimizing the KL divergence between two probability paths, and the KL divergence bound depends on higher-order score functions.

Several works have explored improving training DMs with higher-order score matching. For instance, Meng et al. (2021) have proposed high-order denoising score matching leveraging Tweedie's formula (Robbins, 1992; Efron, 2011) to provide a more accurate local approximation of the data density (e.g., its curvature). We notice that the trace of the second-order score matching proposed in (Meng et al., 2021) resonates with the idea of our proposed FDM in the context of DMs. Additionally, inspired by the KL divergence bound, high-order score matching – matching up to third-order score – has been used to improve likelihood estimation for training DMs (Lu et al., 2022). Nevertheless, these higher-order score-matching methods are significantly more expensive than our proposed FDM.

Enforcing the continuity equation for flow dynamics is another related work that has been studied in the context of DMs. In particular, Lai et al. (2023) shows that the score function satisfies a Fokker-Planck equation (FPE) and directly penalizes the loss function with the error from plug-

ging the learned score function into the score FPE. To the best of our knowledge, developing a PDE characterization of the error between the exact and learned probability paths and bounding their TV gap using only the vector field and its divergence have not been considered in the literature.

## 1.3. Organization

We organize this paper as follows. We provide a brief review of FM in Section 2. In Section 3, we present our theoretical analysis of the gap between the exact and learned probability paths, accompanied by illustrative numerical evidence. We present FDM to improve training FGMs in Section 4. We verify the advantages of FDM over FM using a few representative benchmark tasks in Section 5. Technical proofs, additional experimental details, and experimental results are provided in the appendix.

## 2. Flow Matching

In this section, we provide a brief review of FM and prevalent designs of conditional probability paths. A vector field $\boldsymbol{u}_t : [0, 1] \times \mathbb{R}^d \to \mathbb{R}^d$ defines a flow $\boldsymbol{\psi}_t : [0, 1] \times \mathbb{R}^d \to \mathbb{R}^d$ through the following ODE:

$$\frac{d}{dt}\boldsymbol{\psi}_t(\boldsymbol{x}) = \boldsymbol{u}_t(\boldsymbol{\psi}_t(\boldsymbol{x})), \qquad (3)$$

with the initial condition $\boldsymbol{\psi}_0(\boldsymbol{x}) = \boldsymbol{x}$. FGMs map a prior noise distribution $p_0 = q$ to data distribution $p_1 \approx p$ via the following map:

$$p_t(\boldsymbol{x}) = p_0(\boldsymbol{\psi}_t^{-1}(\boldsymbol{x}))\det\left[\frac{\partial \boldsymbol{\psi}_t^{-1}}{\partial \boldsymbol{x}}(\boldsymbol{x})\right], \ \forall \boldsymbol{x} \in p_0.$$

For a given sample $\boldsymbol{x}_1 \sim p$, FM defines a conditional probability path satisfying $p_0(\boldsymbol{x}|\boldsymbol{x}_1) = q(\boldsymbol{x})$ and $p_1(\boldsymbol{x}|\boldsymbol{x}_1) \approx \delta(\boldsymbol{x} - \boldsymbol{x}_1)$ – the Dirac-delta distribution centered at $\boldsymbol{x}_1$, and the corresponding conditional vector field $\boldsymbol{u}_t(\boldsymbol{x}|\boldsymbol{x}_1)$. Then FM regresses a neural network-parameterized unconditional vector field $\boldsymbol{v}_t(\boldsymbol{x}, \theta)$ by minimizing CFM in equation (2).

A prevalent choice for $p_t(\boldsymbol{x}|\boldsymbol{x}_t)$ is the Gaussian conditional probability path given by

$$p_t(\boldsymbol{x}|\boldsymbol{x}_1) = \mathcal{N}(\boldsymbol{x}|\boldsymbol{\mu}_t(\boldsymbol{x}_1), \sigma_t(\boldsymbol{x}_1)^2\boldsymbol{I}),$$

with $\boldsymbol{\mu}_0(\boldsymbol{x}_1) = \boldsymbol{0}$ and $\sigma_0(\boldsymbol{x}_1) = 1$. Moreover, $\boldsymbol{\mu}_1(\boldsymbol{x}_1) = \boldsymbol{x}_1$ and $\sigma_1(\boldsymbol{x}_1)$ is a small number so that $p_1(\boldsymbol{x}|\boldsymbol{x}_1) \approx \delta(\boldsymbol{x} - \boldsymbol{x}_1)$. Some celebrated DMs can be interpreted as FM models with Gaussian conditional probability paths. In particular, the generation process of the DM with VE SDE (Song et al., 2020) has the conditional probability path:

$$p_t(\boldsymbol{x}|\boldsymbol{x}_1) = \mathcal{N}(\boldsymbol{x}|\boldsymbol{x}_1, \sigma_{1-t}^2\boldsymbol{I}),$$

where $\sigma_t$ is an increasing function satisfying $\sigma_0 = 0$ and $\sigma_1 \gg 1$. The corresponding conditional vector field is given

by

$$\boldsymbol{u}_t(\boldsymbol{x}|\boldsymbol{x}_1) = -\frac{\sigma'_{1-t}}{\sigma_{1-t}}(\boldsymbol{x} - \boldsymbol{x}_1).$$

where $\sigma'_{1-t}$ denote the derivative of the function. Likewise, the VP SDE (Song et al., 2020) has the following conditional probability path:

$$p_t(\boldsymbol{x}|\boldsymbol{x}_1) = \mathcal{N}(\boldsymbol{x}|\alpha_{1-t}\boldsymbol{x}_1, (1 - \alpha_{1-t}^2)\boldsymbol{I}),$$

where $\alpha_t = e^{-\frac{1}{2}T(t)}$ and $T(t) = \int_0^t \beta(s)ds$ with $\beta(s)$ being the noise scale function. The corresponding conditional vector field is

$$\boldsymbol{u}_t(\boldsymbol{x}|\boldsymbol{x}_1) = \frac{\alpha'_{1-t}}{1 - \alpha_{1-t}^2}(\alpha_{1-t}\boldsymbol{x} - \boldsymbol{x}_1).$$

Besides diffusion paths, the optimal transport (OT) path is another remarkable choice (Lipman et al., 2023). OT path uses the Gaussian conditional probability path with

$$\boldsymbol{\mu}_t(\boldsymbol{x}) = t\boldsymbol{x}_1, \text{ and } \sigma_t(\boldsymbol{x}) = 1 - (1 - \sigma_{\min})t.$$

The corresponding conditional vector field is given by

$$\boldsymbol{u}_t(\boldsymbol{x}|\boldsymbol{x}_1) = \frac{\boldsymbol{x}_1 - (1 - \sigma_{\min})\boldsymbol{x}}{1 - (1 - \sigma_{\min})t}.$$

## 3. Error Analysis for Probability Paths

In this section, we analyze the error between the two probability paths associated with the exact and learned vector fields, respectively. Specifically, we show that this error satisfies a PDE similar to the original continuity equation, but with an additional forcing term. Using Duhamel's principle (Seis, 2017), we reveal that this forcing term directly governs the magnitude of the error. The omitted proofs, along with the common assumptions employed by (Lu et al., 2022; Lipman et al., 2023; Albergo et al., 2023) and adopted in our theoretical results, are provided in Appendix A.

### 3.1. PDE for the Error Between Probability Flows

Recall that the marginal probability path $p_t(\boldsymbol{x})$ and the marginal vector field $\boldsymbol{u}_t(\boldsymbol{x})$ satisfy the following continuity equation (Villani et al., 2009):

$$\frac{\partial p_t(\boldsymbol{x})}{\partial t} + \nabla \cdot (p_t(\boldsymbol{x})\boldsymbol{u}_t(\boldsymbol{x})) = 0. \qquad (4)$$

We can rewrite the continuity equation into the following non-conservative form:

$$\frac{\partial p_t(\boldsymbol{x})}{\partial t} = -(\nabla \cdot \boldsymbol{u}_t(\boldsymbol{x}))p_t(\boldsymbol{x}) - \boldsymbol{u}_t(\boldsymbol{x}) \cdot \nabla p_t(\boldsymbol{x}). \quad (5)$$

Similarly, consider the probability path $\hat{p}_t(\boldsymbol{x})$ associated with the neural network-parametrized vector field $\boldsymbol{v}_t(\boldsymbol{x}, \theta)$.

This probability path satisfies the following continuity equation, which has the same initial condition as the ground truth equation (5), i.e., $p_0 = \hat{p}_0$:

$$\frac{\partial \hat{p}_t(\boldsymbol{x})}{\partial t} = -(\nabla \cdot \boldsymbol{v}_t(\boldsymbol{x}, \theta))\hat{p}_t(\boldsymbol{x}) - \boldsymbol{v}_t(\boldsymbol{x}, \theta) \cdot \nabla \hat{p}_t(\boldsymbol{x}). \quad (6)$$

We now introduce the error term $\epsilon_t(\boldsymbol{x}) := p_t(\boldsymbol{x}) - \hat{p}_t(\boldsymbol{x})$. The following proposition shows that $\epsilon_t$ satisfies a PDE similar to equation (4), but with an additional forcing term that reflects the discrepancy between the vector fields $\boldsymbol{u}_t$ and $\boldsymbol{v}_t$.

**Proposition 3.1.** $\epsilon_t := p_t - \hat{p}_t$ *satisfies the following PDE:*

$$\begin{cases} \partial_t \epsilon_t + \nabla \cdot (\epsilon_t \boldsymbol{v}_t) = L_t, \\ \epsilon_0(\boldsymbol{x}) = 0, \end{cases} \quad (7)$$

*where*

$$L_t = -p_t \Big[ \nabla \cdot (\boldsymbol{u}_t - \boldsymbol{v}_t) + (\boldsymbol{u}_t - \boldsymbol{v}_t) \cdot \nabla \log p_t \Big]. \quad (8)$$

### 3.2. Error Bound for Probability Paths

FM aims to minimize the discrepancy between $p_t$ and $\hat{p}_t$ by reducing the difference between their associated vector fields, $\boldsymbol{u}_t(\boldsymbol{x})$ and $\boldsymbol{v}_t(\boldsymbol{x}, \theta)$, through minimizing the CFM loss equation (2). However, Proposition 3.1 highlights that the error dynamics are not only influenced by $\boldsymbol{u}_t - \boldsymbol{v}_t$ but also by $\nabla \cdot (\boldsymbol{u}_t - \boldsymbol{v}_t)$, as both terms contribute to the forcing term in equation (7). To formalize this observation, we solve $\epsilon_t$ using Duhamel's formula (Seis, 2017). In particular, we have the following result:

**Corollary 3.2.** *For any $t \in [0, 1]$, the error $\epsilon_t$ satisfies*

$$\epsilon_t(\phi_t(\boldsymbol{x})) \cdot \det \nabla \phi_t(\boldsymbol{x}) = -\int_0^t L_s(\phi_s(\boldsymbol{x})) \cdot \det \nabla \phi_s(\boldsymbol{x}) ds,$$

*where $\phi_t(\boldsymbol{x})$ is the flow induced by the vector field $\boldsymbol{v}_t(\boldsymbol{x})$ in a similar way as that in equation (3), $\det \nabla \phi_t(\boldsymbol{x})$ denotes the determinant of the Jacobian matrix $\nabla \phi_t(\boldsymbol{x})$, and $L_s$ is defined in Proposition 3.1.*

Corollary 3.2 suggests that minimizing the divergence gap is as important as reducing the vector field discrepancy in order to learn an accurate probability path.

To quantify the error $\epsilon_t$, we consider the following TV distance between $p_t$ and $\hat{p}_t$:

$$\begin{aligned} \mathrm{TV}(p_t, \hat{p}_t) &:= \frac{1}{2} \int \big| p_t(\boldsymbol{x}) - \hat{p}_t(\boldsymbol{x}) \big| d\boldsymbol{x} \\ &= \frac{1}{2} \int \big| \epsilon_t(\boldsymbol{x}) \big| d\boldsymbol{x}. \end{aligned} \quad (9)$$

Motivated by the error-related identity in Corollary 3.2 and the form of $L_t$ in equation (8), we introduce an additional

term as follows:

$$\mathcal{L}_{\mathrm{DM}}(\theta) := \mathbb{E}_{t, p_t(\boldsymbol{x})} \bigg[ \Big| \big| \nabla \cdot (\boldsymbol{u}_t - \boldsymbol{v}_t) + (\boldsymbol{u}_t - \boldsymbol{v}_t) \cdot \nabla \log p_t \big| \Big| \bigg]. \quad (10)$$

The following theorem establishes an upper bound for the error term $\mathrm{TV}(p_t, \hat{p}_t)$ in terms of $\mathcal{L}_{\mathrm{DM}}(\theta)$.

**Theorem 3.3.** *Under some common mild assumptions adopted in (Lu et al., 2022; Lipman et al., 2023; Albergo et al., 2023), the following inequality holds for any $t \in [0, 1]$:*

$$\mathrm{TV}(p_t, \hat{p}_t) \le \frac{1}{2} \mathcal{L}_{\mathrm{DM}}(\theta). \quad (11)$$

*Specifically, $p_t(\boldsymbol{x}) = \hat{p}_t(\boldsymbol{x})$ when $\mathcal{L}_{\mathrm{DM}}$ is zero.*

## 4. Conditional Divergence Matching

In the previous section, we have highlighted the importance of matching the divergence between $\boldsymbol{u}_t$ and $\boldsymbol{v}_t$ beyond matching the vector fields themselves. However, directly minimizing the divergence loss presents a computational challenge, as computing the divergence of the exact unconditional vector field is intractable. To address this issue, we will leverage a similar idea to the conditional flow matching to address the computational issue.

We start by deriving the conditional version of $\mathcal{L}_{\mathrm{DM}}(\theta)$. We recall the following conditional form of the continuity equation from (Lipman et al., 2023):

$$\partial_t p_t(\boldsymbol{x}|\boldsymbol{x}_1) = \nabla \cdot \big( p_t(\boldsymbol{x}|\boldsymbol{x}_1) \boldsymbol{u}_t(\boldsymbol{x}|\boldsymbol{x}_1) \big), \quad (12)$$

which relates the evolution of the conditional probability density $p_t(\boldsymbol{x}|\boldsymbol{x}_1)$ to the divergence of $p_t(\boldsymbol{x}|\boldsymbol{x}_1)\boldsymbol{u}_t(\boldsymbol{x}|\boldsymbol{x}_1)$. By integrating over the conditioning variable $\boldsymbol{x}_1$ and applying the continuity equation (4), we obtain the following connection between the conditional divergence and unconditional divergence:

$$\begin{aligned} &\nabla \cdot \big( p_t(\boldsymbol{x}) \boldsymbol{u}_t(\boldsymbol{x}) \big) \\ =& \partial_t p_t(\boldsymbol{x}) \\ =& \int \partial_t p_t(\boldsymbol{x}|\boldsymbol{x}_1) p(\boldsymbol{x}_1) \, d\boldsymbol{x}_1 \\ =& \int \nabla \cdot \big( p_t(\boldsymbol{x}|\boldsymbol{x}_1) \boldsymbol{u}_t(\boldsymbol{x}|\boldsymbol{x}_1) \big) p(\boldsymbol{x}_1) \, d\boldsymbol{x}_1. \end{aligned} \quad (13)$$

Furthermore, we observe the following identity:

$$\begin{aligned} &p_t \Big[ \nabla \cdot (\boldsymbol{u}_t - \boldsymbol{v}_t) + (\boldsymbol{u}_t - \boldsymbol{v}_t) \cdot \nabla \log p_t \Big] \\ =& \nabla \cdot (p_t \boldsymbol{u}_t) - \nabla \cdot (p_t \boldsymbol{v}_t). \end{aligned}$$

This leads to the following error estimation for the condi-

tional divergence loss:

$$\mathcal{L}_{\text{CDM}}(\theta)$$
$$:= \mathbb{E}_{t,p_t(\boldsymbol{x}|\boldsymbol{x}_1),p(\boldsymbol{x}_1)} \left[ \left| \left( \nabla \cdot \boldsymbol{u}_t(\boldsymbol{x}|\boldsymbol{x}_1) - \nabla \cdot \boldsymbol{v}_t(\boldsymbol{x},\theta) \right) \right. \right.$$
$$\left. \left. + \left( \boldsymbol{u}_t(\boldsymbol{x}|\boldsymbol{x}_1) - \boldsymbol{v}_t(\boldsymbol{x},\theta) \right) \cdot \nabla \log p_t(\boldsymbol{x}|\boldsymbol{x}_1) \right| \right].$$
(14)

Now we are ready to establish the fact that the conditional divergence loss $\mathcal{L}_{\text{CDM}}(\theta)$ is an upper bound for the divergence loss $\mathcal{L}_{\text{DM}}(\theta)$ and the TV gap $\text{TV}(p_t, \hat{p}_t)$. We summary our results in the following theorem:

**Theorem 4.1.** *We have the following inequality:*

$$\mathcal{L}_{\text{DM}}(\theta) \leq \mathcal{L}_{\text{CDM}}(\theta).$$
(15)

*Furthermore, we have:*

$$\text{TV}(p_t, \hat{p}_t) \leq \frac{1}{2}\mathcal{L}_{\text{CDM}}(\theta),$$
(16)

*for any $t \in [0, 1]$.*

### 4.1. Flow and Divergence Matching.

In practice, we observe that minimizing $\mathcal{L}_{\text{CDM}}(\theta)$ alone cannot yield appealing results, as the loss cannot go to exact zero in training. This nonzero loss comes from a balance between $\nabla \cdot \boldsymbol{u}_t(\boldsymbol{x}|\boldsymbol{x}_1) - \nabla \cdot \boldsymbol{v}_t(\boldsymbol{x},\theta)$ and $(\boldsymbol{u}_t(\boldsymbol{x}|\boldsymbol{x}_1) - \boldsymbol{v}_t(\boldsymbol{x},\theta)) \cdot \nabla \log p_t(\boldsymbol{x}|\boldsymbol{x}_1)$, and both terms can be positive or negative, resulting in cancellation. As such, there is no guarantee that we can learn a vector field $\boldsymbol{v}_t(\boldsymbol{x},\theta)$ that is in proximity to $\boldsymbol{u}_t(\boldsymbol{x})$ by minimizing $\mathcal{L}_{\text{CDM}}(\theta)$. In contrast, by using a weighted sum of $\mathcal{L}_{\text{CDM}}$ and $\mathcal{L}_{\text{CFM}}$ as the training objective, we can directly control the gap between the vector fields and their divergences. Therefore, we propose the flow and divergence matching (FDM) loss:

$$\mathcal{L}_{\text{FDM}} = \lambda_1 \mathcal{L}_{\text{CFM}} + \lambda_2 \mathcal{L}_{\text{CDM}},$$
(17)

where $\lambda_1, \lambda_2 > 0$ are hyperparameters; we choose them via hyperparameter search in this work. It is an interesting future direction to design a principle to choose $\lambda$s optimally.

*Remark* 4.2. It is worth noting that minimizing the objective function $\mathcal{L}_{\text{FDM}}$ offers a more computationally efficient approach compared to higher-order control methods presented in (Lu et al., 2022; Lai et al., 2023), as it is computationally much cheaper than controlling differences in higher-order quantities (e.g., gradient of the divergence). Moreover, to further improve training efficiency, we introduce an efficient squared conditional divergence-matching loss $\mathcal{L}_{\text{CDM-2}}^{\text{eff}}$, which adopts stop-gradient (Lu et al., 2022) and Hutchinson trace estimation (Hutchinson, 1989) techniques. This adds only one extra backward pass compared with baseline flow-matching training; see Appendix D for details. While a bounded TV distance does not necessarily imply a bound on

the KL divergence, we leave the exploration of developing a computationally efficient method for controlling the KL divergence in this direction for future work.

### 4.2. Synthetic Experiment.

To solidify our theoretical results, we present a simple numerical example before moving to real-world applications. Specifically, we consider the problem of sampling from the following Gaussian mixture distribution

$$p(\boldsymbol{x}) = 0.23\mathcal{N}(-3, 0.1) + 0.35\mathcal{N}(-1, 0.1)$$
$$+ 0.15\mathcal{N}(-1, 0.1) + 0.27\mathcal{N}(3, 0.1),$$
(18)

using both standard FM and our proposed FDM defined in equation (17). We use a 3-layer MLP to approximate the VP diffusion path vector field by minimizing equation (17) with $\lambda_1 = 1, \lambda_2 = 0$ for FM and $\lambda_1 = 1, \lambda_2 = 0.2$ for FDM. We use $10^4$ data points sampled from equation (18) for training.

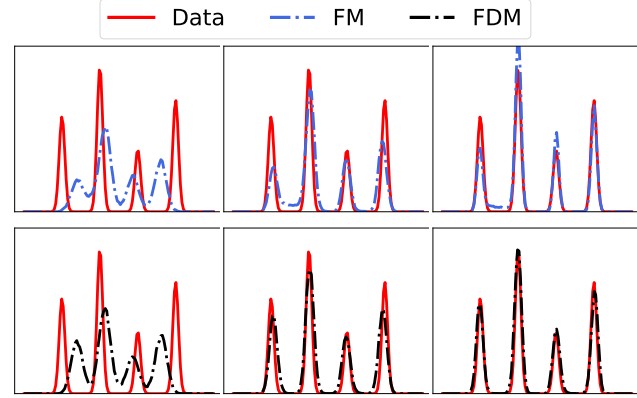

*Figure 2.* Snapshots for probability paths at $t = 0.6, 0.85$, and 1 (left to right). First/Second row: FM/FDM vs. data distribution.

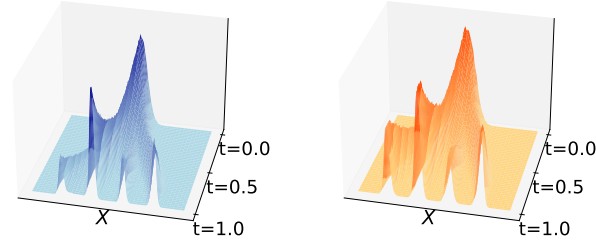

*Figure 3.* Comparison of probability paths over time learned by FM (left) vs. FDM (right).

Figures 2 and 3 contrast the performance of our proposed FDM against the baseline FM. The numerical results confirm that the probability path (at $t = 1$) learned by FM suffers from a substantial discrepancy from the exact Gaussian mixture distribution. In contrast, FDM learns the Gaussian

mixture much more accurately than FM. Specifically, the TV gaps between the learned and exact distributions are 0.0945 and 0.0587 for FM and FDM, respectively.

# 5. Experimental Results

In this section, we validate the efficacy and efficiency of the proposed FDM in enhancing FM across various benchmark tasks, including density estimation on synthetic 2D data (Section 5.1.1) and image data (Section 5.1.2), DNA sequence generation (Section 5.2), and spatiotemporal data sampling tasks including trajectory sampling for dynamical systems (Section 5.3.1) and video prediction via latent FM (Section 5.3.2).

Our experiments confirm that our proposed FDM remarkably improves FM with guidance, enhancing promoter DNA sequence design with class-conditional flow, as well as refining trajectory generation for dynamical systems and video predictions conditioning on the initial states over the first several time steps. In this section, we report the error between the exact and learned distributions in terms of the TV distance, and the corresponding KL divergence results are further provided in Appendix C.

**Software and Equipment.** Our implementation utilizes PyTorch Lightning (Falcon, 2019) for synthetic density estimation, DNA sequence generation, and video generation, while JAX (Bradbury et al., 2018) and TensorFlow (Abadi et al., 2016) are employed for dynamical systems-related experiments. Experiments are conducted on multiple NVIDIA RTX 3090 GPUs.

**Training Setup.** See Appendix B.

**Models and Datasets.** We employ OT and VE/VP diffusion paths for the flow maps in most tasks except the Dirichlet flow for DNA generation. We follow the approach used in (Huang et al., 2024; Lu et al., 2022) to estimate the divergence, which employs Hutchinson's trace estimator (Hutchinson, 1989). Our experiments utilize a numerical simulation-based dataset for density estimation and trajectory sampling, a dataset extracted from a database of human promoters (Hon et al., 2017) for DNA design, and the KTH human motion dataset (Schuldt et al., 2004) and the BAIR Robot Pushing dataset (Ebert et al., 2017) for video prediction.

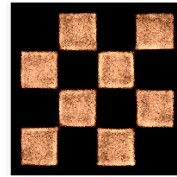 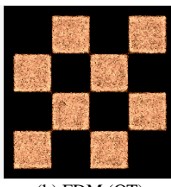 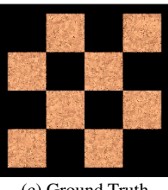

| (a) FM (OT) | (b) FDM (OT) | (c) Ground Truth |

*Figure 4.* Generated samples from FM and FDM using the optimal transport (OT) path trained on the checkerboard dataset.

## 5.1. Density Estimation on Synthetic and Image Data

We train the models for density estimation on two datasets: a synthetic 2D checkerboard and the image dataset CIFAR-10 (Krizhevsky et al., 2009).

### 5.1.1. SYNTHETIC DENSITY ESTIMATION

In this experiment, we train models using FM and FDM for $2 \times 10^4$ iterations using a batch size of 512. For each iteration, we numerically sample the data for the training set and use the same sampling method for validation and testing sets. We compare FM and FDM with the baselines for both OT and VP paths in the likelihood computed based on the test dataset. The results in Table 1 and Fig. 4 show that FDM consistently outperforms FM across different probability paths.

### 5.1.2. DENSITY MODELING ON IMAGE DATASETS

In the experiment, we train models using both FM and FDM for image sampling on the CIFAR10 dataset (Krizhevsky et al., 2009). We follow the experimental settings in the flow matching baseline paper (Lipman et al., 2023) and compare the performance in terms of the negative log-likelihood and FID scores of the sampled images as shown in Table 2.

| Model | NLL($\downarrow$) | FID($\downarrow$) |
|---|---|---|
| FM(OT) | 2.99 | 6.35 |
| FDM(OT) | 2.85 | 5.62 |

*Table 2.* Negative log-likelihood and sample quality (FID scores) estimation on CIFAR-10.

## 5.2. Sequential Data Sampling–DNA Sequence

In this experiment, we demonstrate that FDM enhances FM with the conditional OT path and Dirichlet path (Stark et al., 2024) on the probability simplex for DNA sequence generation, both with and without guidance, following experiments conducted in (Stark et al., 2024). For this task, instead of directly parameterizing the vector field, the Dirichlet flow model constructs it by combining pre-designed Dirichlet probability path functions with a parameterized classifier $\hat{p}_t(\boldsymbol{x}_1|\boldsymbol{x},\theta)$, where $\boldsymbol{x}$ is sampled from the conditional proba-

| Model | FM (OT) | FDM (OT) | FM (VP) | FDM (VP) |
|---|---|---|---|---|
| Likelihood ($\uparrow$) | $2.38_{\pm.02}$ | $\mathbf{2.53}_{\pm.02}$ | $2.34_{\pm.02}$ | $2.46_{\pm.02}$ |

*Table 1.* Likelihood estimation of models on the checkerboard test set. Here, "OT" denotes the optimal transport path and "VP" denotes the variance-preserving path. Unit: $\times 10^{-2}$

bility at time $t$, given the data point $\boldsymbol{x}_1$. Since $\boldsymbol{x}_1$ represents discrete categorical data with a finite number of categories, it can be treated as a class label. Since this approach only requires parameterizing the classifier $\hat{p}_t(\boldsymbol{x}_1|\boldsymbol{x}, \theta)$, we only need to penalize the norm of the gradient with respect to the input of the classifier, which is equivalent to minimizing the divergence error; see Appendix B.2 for more details. Additionally, we conduct experiments where the classifier is parameterized with guidance, $p_t(\boldsymbol{x}_1|\boldsymbol{x}, \boldsymbol{y}, \theta)$ with $\boldsymbol{y}$ representing the guiding information. We use the same experiment setup in (Stark et al., 2024) except the newly introduced hyperparameters $\lambda_1$ and $\lambda_2$; see Appendix B.2 for the detailed settings.

### 5.2.1. SIMPLEX DIMENSION WITHOUT GUIDANCE

We first evaluate the performance of FM and FDM in a non-guided simple generation task. The data is sampled from a uniform Dirichlet distribution with a sequence length of $l = 4$ and $K = 40$ categories. We compare the TV distance and KL divergence between the generated distribution and the target distribution on the test dataset. The results in Table 3 and Table 14 in Appendix C.1 demonstrate that FDM outperforms FM in generating the simple sequential categorical data.

| Method | TV Distance | Time (s/iter) |
| --- | --- | --- |
| Linear FM | $0.12_{\pm 0.005}$ | 0.10 |
| Linear FDM | $0.10_{\pm 0.004}$ | 0.16 |
| Dirichlet FM | $0.08_{\pm 0.005}$ | 0.10 |
| Dirichlet FDM | $\mathbf{0.07}_{\pm 0.004}$ | 0.16 |

*Table 3.* TV distances between the generated and target distributions.

### 5.2.2. PROMOTER DNA SEQUENCE DESIGN WITH GUIDANCE

We further evaluate the ability of FM and FDM in training generative models for designing DNA promoter sequences guided by a desired promoter profile. We train the models guided by a profile by providing it as additional input to the vector field and evaluate generated sequences using mean-squared error (MSE) between their predicted and original regulatory activity, as determined by SEI (Chen et al., 2022). We include the discrete DM (Albergo et al., 2023) and the language model (Stark et al., 2024) for comparison in Table 4. For this task, we use a dataset of 100,000 promoter sequences with 1024 base pairs extracted from a database of human promoters (Hon et al., 2017). See Appendix B.2 for more details about the dataset. The results confirm that FDM improves FM in training guided models for categorical data generation.

| Method | MSE ($\downarrow$) |
| --- | --- |
| Bit Diffusion (One-hot Encoding)(Albergo et al., 2023) | 3.95E-2 |
| DDSM (Albergo et al., 2023) | 3.34E-2 |
| Large Language Model (Stark et al., 2024) | 3.33E-2 |
| Linear FM (Stark et al., 2024) | $2.82_{\pm 0.02}$E-2 |
| Linear FDM (ours) | $2.78_{\pm 0.01}$E-2 |
| Dirichlet FM (Stark et al., 2024) | $2.68_{\pm 0.01}$E-2 |
| Dirichlet FDM (ours) | $\mathbf{2.59}_{\pm 0.02}$E-2 |

*Table 4.* Evaluation of transcription profile guided promoter DNA sequence design of different models.

### 5.3. Spatiotemperal Data Generation

In this section, we evaluate our model on spatiotemporal data sampling tasks, both with and without guidance. Specifically, we consider two scenarios: trajectory sampling for dynamical systems and video generation.

### 5.3.1. TRAJECTORY SAMPLING FOR DYNAMICAL SYSTEMS

Sampling trajectories for dynamical systems under event guidance is crucial for understanding and predicting the climate and beyond (Perkins & Alexander, 2013; Mosavi et al., 2018; Hochman et al., 2019). Finzi et al. (2023) develop a DM for sampling these events.

In this experiment, we compare FDM against FM and DM from (Finzi et al., 2023) on the Lorenz and FitzHugh-Nagumo dynamical systems (Farazmand & Sapsis, 2019); the details of these systems are provided in Appendix B.1. We test sampling trajectories from these systems with and without event guidance. A trajectory, either from a dataset or sampled, is a discrete time series of vectors concatenated into $\boldsymbol{x}_1 = [\boldsymbol{x}(\tau_m)]_{m=1}^{M} \in \mathbb{R}^{Md}$, where $M$ is the number of time steps and $d$ is the dimension of the system. Following (Finzi et al., 2023), an event $E$ is a set of trajectories characterized by some event constraint; for example, $E = \{\boldsymbol{x}_1 : C(\boldsymbol{x}_1) > 0\}$, where the event constraint function $C : \mathbb{R}^{Md} \to \mathbb{R}$ is smooth. The challenge of this experiment is to sample trajectories in $E$ when $C$ is only known after the models have been trained. The detailed sampling procedure using DM can be found in (Finzi et al., 2023).

The event-guided sampling procedure from (Finzi et al., 2023) uses Tweedie's formula (Robbins, 1992; Efron, 2011), which requires the score function $\nabla \log p_t(\boldsymbol{x})$. Since FM and FDM are not trained to approximate $\nabla \log p_t(\boldsymbol{x})$ directly, we derive an approximation formula using the learned vector field $\boldsymbol{v}_t(\boldsymbol{x}, \theta)$. Applying Lemma 1 of (Lipman et al., 2023) to the probability flow ODE (Song et al., 2020), the evolution of $p_t(\boldsymbol{x})$ satisfies:

$$\boldsymbol{u}_t(\boldsymbol{x}) = -\boldsymbol{f}(\boldsymbol{x}, 1-t) + \frac{1}{2}g^2(1-t)\nabla \log p_{1-t}(\boldsymbol{x}), \quad (19)$$

where $\boldsymbol{f}$ is the drift term and $g$ is the noise coefficient.

Rearranging equation (19), we express $\nabla \log p_t(\boldsymbol{x})$ in terms of $\boldsymbol{u}_t(\boldsymbol{x})$, then approximate $\boldsymbol{u}_t(\boldsymbol{x})$ by $\boldsymbol{v}_t(\boldsymbol{x}, \theta)$.

We use the events defined in (Finzi et al., 2023) for our experiments. The event for the Lorenz system is when a trajectory stays on one arm of the chaotic attractor. This is characterized by $C(\boldsymbol{x}) = 0.6 - \|F[\boldsymbol{x} - \overline{\boldsymbol{x}}]\|_1 > 0$, where $F$ is the Fourier transform over trajectory time $\tau$, $\|\cdot\|_1$ is the 1-norm summing over both over the frequency magnitudes and the three dimensions of $\boldsymbol{x}(\tau)$, and $\overline{\boldsymbol{x}}$ is the average of $\boldsymbol{x}(\tau)$ over $\tau$. For the FitzHugh-Nagumo system, the event is neuron spiking, which is characterized by $C(\boldsymbol{x}) = \max_\tau[x_1(\tau) + x_2(\tau)]/2 - 2.5 > 0$.

We compare the models' ability to generate trajectories according to $p(\boldsymbol{x}_1)$ and $p(\boldsymbol{x}_1|E)$ by computing a test set of trajectories using the Dormand-Prince ODE solver (Dormand & Prince, 1980) and sampling trajectories using each model. Table 5 presents the TV distance between the model and the distributions. From the result, we observe that FDM achieves the lowest TV distance for every distribution. The TV distance of FDM is smaller than that of FM, which empirically demonstrates that the divergence mismatch has a significant effect on the error $\epsilon_t(\boldsymbol{x}_t)$.

Furthermore, this shows that the proposed loss $\mathcal{L}_{\text{FDM}}$ effectively reduces the mismatch. This mismatch reduction also enables FDM to attain the lowest negative log-likelihood (NLL) estimates. Table 6 shows the mean NLL over trajectories and trajectory dimension with respect to to $p(\boldsymbol{x}_1)$, while Fig. 5 compares the histograms of the event constraint value of each event trajectory. Importantly, these improvements of FDM do not trade off with its accuracy in estimating $p(E)$. When $p(E)$ is estimated based on the proportion of sampled trajectories that fall within $E$, all the models are comparable. Table 7 reports the KL divergence between the histograms of the event constraint value $C(\boldsymbol{x}_1)$ for event trajectories $\boldsymbol{x}_1$ from the dataset computed by an ODE solver and those sampled with event guidance from the models. The results show that our FDM consistently outperforms both the FM and Diffusion models.

| Model | Lorenz | | FitzHugh-Nagumo | |
|---|---|---|---|---|
| | $p(\boldsymbol{x}_1)$ ($\downarrow$) | $p(\boldsymbol{x}_1\|E)$ ($\downarrow$) | $p(\boldsymbol{x}_1)$ ($\downarrow$) | $p(\boldsymbol{x}_1\|E)$ ($\downarrow$) |
| Diffusion | 0.0314 | 0.1001 | 0.0277 | 0.1192 |
| FM | 0.0348 | 0.0972 | 0.0314 | 0.2164 |
| FDM(ours) | **0.0306** | **0.0914** | **0.0266** | **0.1168** |

*Table 5.* TV distances of the models from the trajectory distribution $p(\boldsymbol{x}_1)$ and from the distribution conditioned on an event $p(\boldsymbol{x}_1|E)$. Here, Diffusion results follow from (Finzi et al., 2023), while FM and FDM are based on our implementation, which builds on the code provided by Finzi et al. (2023).

### 5.3.2. GENERATIVE MODELING FOR VIDEOS

We aim to show how FDM pushes the boundary of FM performance for sequential data generation in a latent space.

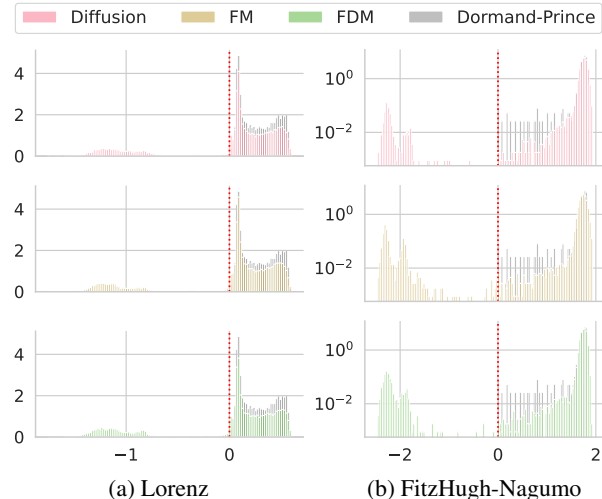

(a) Lorenz  (b) FitzHugh-Nagumo

*Figure 5.* Histograms of the constraint value $C(\boldsymbol{x}_1)$ where $\boldsymbol{x}_1$ is an event trajectory computed by the Dormand-Prince ODE solver or sampled from the model with event guidance. The unguided sampling histograms are shown in Appendix C.2.

| Model | Lorenz | | FitzHugh-Nagumo | |
|---|---|---|---|---|
| | NLL($\boldsymbol{x}_1$) ($\downarrow$) | $p(E)$ | NLL($\boldsymbol{x}_1$) ($\downarrow$) | $p(E)$ |
| Dormand-Prince | – | 0.197 | – | 0.035 |
| Diffusion | -7.052 | 0.200 | -7.365 | 0.032 |
| FM | -13.190 | 0.199 | -13.942 | 0.034 |
| FDM | **-14.361** | 0.200 | **-14.408** | 0.033 |

*Table 6.* NLLs averaged over trajectories and trajectory dimension with respect to the trajectory distribution $p(\boldsymbol{x}_1)$, and the likelihood of the user-defined event estimated by the proportion of trajectories contained in event $E$ sampled from the model without guidance. Here, the Diffusion follows from (Finzi et al., 2023). FM and FDM are based on our own implementation.

| Model | Lorenz | | FitzHugh-Nagumo | |
|---|---|---|---|---|
| | $p(\boldsymbol{x}_1)$ | $p(\boldsymbol{x}_1\|E)$ | $p(\boldsymbol{x}_1)$ | $p(\boldsymbol{x}_1\|E)$ |
| Diffusion | 0.0056 | 0.2774 | 0.0260 | 0.3011 |
| FM | 0.0081 | **0.2560** | 0.0280 | 0.3468 |
| FDM(ours) | **0.0049** | 0.3045 | **0.0280** | **0.2084** |

*Table 7.* KL divergence between the histograms of the event constraint value $C(\boldsymbol{x}_1)$ for event trajectories $\boldsymbol{x}_1$ in the dataset of trajectories computed by an ODE solver and event trajectories sampled with event guidance from the models.

We train a latent FM (Davtyan et al., 2023) and a latent FDM for video prediction. We utilize a pre-trained VQGAN (Esser et al., 2021) to encode (resp. decode) each frame of the video to (resp. from) the latent space. We train the models using the latent state at $t-1$ and $t-\tau$, where $\tau$ is randomly selected from $\{2, \ldots, t\}$, by providing them as additional input guidance to the vector field at $t > C$, where $C$ is a positive integer. At inference time, we use the frames at time $t = 0$ to $t = C$ of a video as the guidance and then utilize flow matching to predict the frames after $t = C$.

We consider the human motion dataset – KTH (Schuldt et al., 2004) and BAIR Robot Pushing dataset (Ebert et al., 2017). We follow the experimental setup of (Davtyan et al., 2023); see Appendix B.3 for details. To evaluate the generated samples, we compute the Fréchet video distance (FVD) (Unterthiner et al., 2018) and peak signal-to-noise ratio (PSNR) (Huynh-Thu & Ghanbari, 2008).

**KTH Dataset:** For KTH, we use the first 10 frames as guidance and predict the next 30 frames. The results in Table 8 indicate that FDM enhances latent FM for temporal data generation. Furthermore, Fig. 6 presents illustrative cases showing that our FDM consistently maintains high visual quality throughout the video, whereas the FM model exhibits noticeable degradation in later frames, including loss of fine motion details, missing body parts, and motion failure.

**BAIR Dataset:** For BAIR, we predict 15 future frames based on a single initial frame, with each frame having a resolution of $64 \times 64$ pixels. Because of the highly stochastic motion in the BAIR dataset, following (Davtyan et al., 2023), we generate 100 samples per test video – each conditioned on the same initial frame – and compute metrics over $100 \times 256$ generated samples against 256 randomly selected test videos. To highlight the effectiveness of FDM, we omit the frame refinement step used in (Davtyan et al., 2023). As mentioned in (Davtyan et al., 2023), many models for the BAIR task are computationally expensive, whereas latent FM achieves a favorable trade-off between FVD and computational cost. Our approach further improves latent FM with acceptable additional computational overhead, as shown in Table 9.

We notice that the experiments in Chen et al. (2024) achieve very impressive results for video generation, and it is an interesting future direction to integrate our approach into their framework.

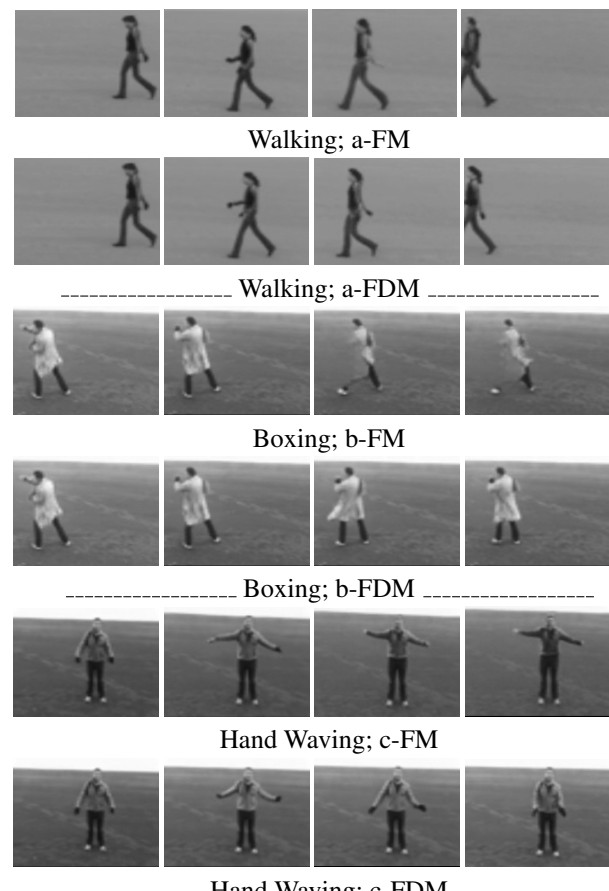

*Figure 6.* Samples on KTH human motion dataset – at frame 0, 13, 27, 40 from left to right – generated by latent FM (a-FM, b-FM, c-FM) and latent FDM (a-FDM, b-FDM, c-FDM).

## 6. Concluding Remarks

In this paper, we have developed a new upper bound for the gap between learned and ground-truth probability paths using FM. Our new error bound shows that FM can be improved by ensuring the divergences of the vector fields are in proximity. To achieve this, we derive a new conditional divergence loss with computational efficiency. Our new training approach – flow and divergence matching – significantly improves FM on various challenging tasks. There are several avenues for future work. A particularly intriguing direction is to develop a computationally efficient method for controlling the KL divergence, for example by integrating deep equilibrium models (Bai et al., 2019) into our framework – similar to how prior works have incorporated them into diffusion (score-based) models (Huang et al., 2024; Bai & Melas-Kyriazi, 2024). This remains an open problem and an important avenue for future research. Moreover, exploring our approach in the Schrödinger bridge setting (Tong et al., 2024) is also an interesting problem.

| Method | FVD(↓) | PSNR(↑) | Time(s/iter) |
|---|---|---|---|
| SRVP (Franceschi et al., 2020) | 222 | 29.7 | – |
| SLAMP (Akan et al., 2021) | 228 | 29.4 | – |
| Latent FM (Davtyan et al., 2023) | 180 | 30.4 | 0.18 |
| Latent FDM (ours) | **155.5**$_{\pm 5}$ | **31.2** | 0.27 |

*Table 8.* KTH dataset evaluation. The evaluation protocol is to predict the next 30 frames given the first 10 frames.

| Method | FVD(↓) | MEM(GB) | Time(hours) |
|---|---|---|---|
| TriVD-GAN-FP (Luc et al., 2020) | 103 | 1024 | 280 |
| Video Transformer (Weissenborn et al., 2019) | 94 | 512 | 336 |
| LVT (Rakhimov et al., 2020) | 126 | 128 | 48 |
| RaMViD (Diffusion) (Höppe et al., 2022) | 84 | 320 | 72 |
| Latent FM (Davtyan et al., 2023) | 146 | 24.2 | 25 |
| Latent FDM (ours) | **123**$_{\pm 4.5}$ | 35 | 36 |

*Table 9.* BAIR dataset evaluation. We adopt the standard evaluation setup, where the model predicts 15 future frames conditioned on a single initial frame. MEM stands for peak memory footprint.

## Acknowledgement

This material is based on research sponsored by NSF grants DMS-2152762, DMS-2208361, DMS-2219956, and DMS-2436344, and DOE grants DE-SC0023490, DE-SC0025589, and DE-SC0025801. HZ acknowledges the support from the U.S. Department of Energy, Office of Science, Office of Advanced Scientific Computing Research research grant DOE-FOA-2493 "Data-intensive scientific machine learning", under contract DE-AC02-06CH11357 at Argonne National Laboratory.

## Impact Statement

This work presents a new theoretical bound on the gap between the exact and learned probability paths using flow matching. The new theoretical bound informs the design of a new efficient training objective to improve flow matching. Our work directly contributes to advancing flow-based generative modeling. Flow-based models have achieved remarkable results in climate modeling and molecular modeling. By developing new theoretical understandings and fundamental algorithms with performance guarantees, we expect our work will advance climate modeling and molecular sciences using generative models. Our work contributes to basic research, and we do not see potential ethical concerns or negative societal impact beyond the current AI.

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

# Appendix for
## *Improving Flow Matching by Aligning Flow Divergence*

## A. Missing Proofs

**Proposition 3.1.** $\epsilon_t := p_t - \hat{p}_t$ *satisfies the following PDE:*

$$\begin{cases} \partial_t \epsilon_t + \nabla \cdot \left( \epsilon_t \boldsymbol{v}_t \right) = L_t, \\ \epsilon_0(\boldsymbol{x}) = 0, \end{cases} \tag{7}$$

*where*

$$L_t = -p_t \Big[ \nabla \cdot (\boldsymbol{u}_t - \boldsymbol{v}_t) + (\boldsymbol{u}_t - \boldsymbol{v}_t) \cdot \nabla \log p_t \Big]. \tag{8}$$

*Proof of Proposition 3.1.* For simplicity, we denote $\frac{\partial}{\partial t}$ by $\partial_t$. From the continuity equations 5 and 6, we have:

$$\begin{aligned} \partial_t \epsilon_t &= \partial_t p_t - \partial_t \hat{p}_t \\ &= \Big[ -p_t (\nabla \cdot \boldsymbol{u}_t) - \boldsymbol{u}_t \cdot \nabla p_t \Big] - \Big[ -\hat{p}_t (\nabla \cdot \boldsymbol{v}_t) - \boldsymbol{v}_t \cdot \nabla \hat{p}_t \Big] \\ &= -p_t (\nabla \cdot \boldsymbol{u}_t) + \hat{p}_t (\nabla \cdot \boldsymbol{v}_t) - \boldsymbol{u}_t \cdot \nabla p_t + \boldsymbol{v}_t \cdot \nabla \hat{p}_t \\ &= -p_t \big( \nabla \cdot (\boldsymbol{u}_t - \boldsymbol{v}_t) \big) - (\nabla \cdot \boldsymbol{v}_t) \epsilon_t - (\boldsymbol{u}_t - \boldsymbol{v}_t) \cdot \nabla p_t - \boldsymbol{v}_t \cdot \nabla \epsilon_t \end{aligned} \tag{20}$$

Rewriting it, we find:

$$\partial_t \epsilon_t + \nabla \cdot (\epsilon_t \boldsymbol{v}_t) = -p_t \big( \nabla \cdot (\boldsymbol{u}_t - \boldsymbol{v}_t) \big) - p_t (\boldsymbol{u}_t - \boldsymbol{v}_t) \cdot \nabla \log p_t \tag{21}$$

Let us define $L_t := -p_t \big( \nabla \cdot (\boldsymbol{u}_t - \boldsymbol{v}_t) \big) - p_t (\boldsymbol{u}_t - \boldsymbol{v}_t) \cdot \nabla \log p_t$. This gives the following PDE for $\epsilon_t$ with the initial condition $\epsilon_0 = p_0 - \hat{p}_0 = 0$:

$$\begin{cases} \partial_t \epsilon_t + \nabla \cdot \left( \epsilon_t \boldsymbol{v}_t \right) = L_t, \\ \epsilon_0(\boldsymbol{x}) = 0. \end{cases} \tag{22}$$

$\square$

**Corollary 3.2.** *For any $t \in [0,1]$, the error $\epsilon_t$ satisfies*

$$\epsilon_t(\phi_t(\boldsymbol{x})) \cdot \det \nabla \phi_t(\boldsymbol{x}) = - \int_0^t L_s(\phi_s(\boldsymbol{x})) \cdot \det \nabla \phi_s(\boldsymbol{x}) ds,$$

*where $\phi_t(\boldsymbol{x})$ is the flow induced by the vector field $\boldsymbol{v}_t(\boldsymbol{x})$ in a similar way as that in equation (3), $\det \nabla \phi_t(\boldsymbol{x})$ denotes the determinant of the Jacobian matrix $\nabla \phi_t(\boldsymbol{x})$, and $L_s$ is defined in Proposition 3.1.*

*Proof of Corollary 3.2.* Let $\phi_t$ denote the flow of the vector field $\boldsymbol{v}_t$, i.e.

$$\begin{cases} \partial_t \phi_t = \boldsymbol{v}_t \big( \phi_t(\boldsymbol{x}) \big), \\ \phi_0(\boldsymbol{x}) = \boldsymbol{x}. \end{cases} \tag{23}$$

Using Duhamel's formula (refer to (Seis, 2017)), we have the following formula for $\epsilon_t$:

$$\epsilon_t \big( \phi_t(\boldsymbol{x}) \big) \det \nabla \phi_t(\boldsymbol{x}) \quad = \epsilon_0(\boldsymbol{x}) + \int_0^t L_s \big( \phi_s(\boldsymbol{x}) \big) \det \nabla \phi_s(\boldsymbol{x}) \, ds = \int_0^t L_s \big( \phi_s(\boldsymbol{x}) \big) \det \nabla \phi_s(\boldsymbol{x}) \, ds \tag{24}$$

$\square$

**Theorem 3.3.** *Under some common mild assumptions adopted in (Lu et al., 2022; Lipman et al., 2023; Albergo et al., 2023), the following inequality holds for any $t \in [0,1]$:*

$$\mathrm{TV}(p_t, \hat{p}_t) \le \frac{1}{2} \mathcal{L}_{\mathrm{DM}}(\theta). \tag{11}$$

*Specifically, $p_t(\boldsymbol{x}) = \hat{p}_t(\boldsymbol{x})$ when $\mathcal{L}_{\mathrm{DM}}$ is zero.*

*Proof of Theorem 3.3.* Note that the total variation distance is defined as:

$$\mathrm{TV}(p_t, \hat{p}_t) = \frac{1}{2} \int \left| p_t(\boldsymbol{x}) - \hat{p}_t(\boldsymbol{x}) \right| d\boldsymbol{x} = \frac{1}{2} \int \left| \epsilon_t(\boldsymbol{x}) \right| d\boldsymbol{x} \tag{25}$$

Using the change of variables twice and applying the formula in Corollary 3.2, we obtain:

$$\begin{aligned}
\mathrm{TV}(p_t, \hat{p}_t) &= \frac{1}{2} \int \left| \epsilon_t(\boldsymbol{x}) \right| d\boldsymbol{x} \\
&= \frac{1}{2} \int \left| \epsilon_t(\phi_t(\boldsymbol{x})) \right| d\,\phi_t(\boldsymbol{x}) \\
&= \frac{1}{2} \int \left| \epsilon_t(\phi_t(\boldsymbol{x})) \det \nabla \phi_t(\boldsymbol{x}) \right| d\boldsymbol{x} \\
&= \frac{1}{2} \int \left| \int_0^t L_s(\phi_s(\boldsymbol{x})) \det \nabla \phi_s(\boldsymbol{x})\, ds \right| d\boldsymbol{x} \\
&\leq \frac{1}{2} \int \int_0^t \left| L_s(\phi_s(\boldsymbol{x})) \det \nabla \phi_s(\boldsymbol{x}) \right| ds\, d\boldsymbol{x} \\
&= \frac{1}{2} \int_0^t \int \left| L_s(\boldsymbol{x}) \right| d\boldsymbol{x}\, ds
\end{aligned} \tag{26}$$

Substituting the expression for $L_s$, we see that:

$$\begin{aligned}
2\,\mathrm{TV}(p_t, \hat{p}_t) &\leq \int_0^t \int \left| p_t(\nabla \cdot (\boldsymbol{u}_s - \boldsymbol{v}_s)) + p_s(\boldsymbol{u}_s - \boldsymbol{v}_s) \cdot \nabla \log p_s \right| d\boldsymbol{x}\, ds \\
&\leq \int_0^t \mathbb{E}_{p_s} \left| \nabla \cdot (\boldsymbol{u}_s - \boldsymbol{v}_s) + (\boldsymbol{u}_s - \boldsymbol{v}_s) \cdot \nabla \log p_s \right| ds \\
&\leq \int_0^T \mathbb{E}_{p_t} \left| \nabla \cdot (\boldsymbol{u}_t - \boldsymbol{v}_t) + (\boldsymbol{u}_t - \boldsymbol{v}_t) \cdot \nabla \log p_t \right| dt \\
&= \mathcal{L}_{\mathrm{DM}}(\theta).
\end{aligned} \tag{27}$$

This completes the proof. $\qquad\square$

**Theorem 4.1.** *We have the following inequality:*

$$\mathcal{L}_{\mathrm{DM}}(\theta) \leq \mathcal{L}_{\mathrm{CDM}}(\theta). \tag{15}$$

*Furthermore, we have:*

$$\mathrm{TV}(p_t, \hat{p}_t) \leq \frac{1}{2} \mathcal{L}_{\mathrm{CDM}}(\theta), \tag{16}$$

*for any $t \in [0, 1]$.*

*Proof of Theorem 4.1.* From equation (13), we can show that:

$$\begin{aligned}
p_t(\nabla \cdot \boldsymbol{u}_t + \boldsymbol{u}_t \cdot \nabla \log p_t) &= \nabla \cdot (p_t \boldsymbol{u}_t) \\
&= \int \nabla \cdot (p_t(\boldsymbol{x}|\boldsymbol{x}_1) \boldsymbol{u}_t(\boldsymbol{x}|\boldsymbol{x}_1)) p(\boldsymbol{x}_1)\, d\boldsymbol{x}_1 \\
&= \int \Big( p_t(\boldsymbol{x}|\boldsymbol{x}_1) \nabla \cdot \boldsymbol{u}_t(\boldsymbol{x}|\boldsymbol{x}_1) + \boldsymbol{u}_t(\boldsymbol{x}|\boldsymbol{x}_1) \cdot \nabla p_t(\boldsymbol{x}|\boldsymbol{x}_1) \Big) p(\boldsymbol{x}_1)\, d\boldsymbol{x}_1.
\end{aligned} \tag{28}$$

On the other hand, we have

$$
\begin{aligned}
p_t\big(\nabla \cdot \boldsymbol{v}_t + \boldsymbol{v}_t \cdot \nabla \log p_t\big) &= p_t \nabla \cdot \boldsymbol{v}_t + \boldsymbol{v}_t \cdot \nabla p_t \\
&= \left( \int p_t(\boldsymbol{x}|\boldsymbol{x}_1) p(\boldsymbol{x}_1)\, d\boldsymbol{x}_1 \right) \nabla \cdot \boldsymbol{v}_t + \boldsymbol{v}_t \cdot \nabla \left( \int p_t(\boldsymbol{x}|\boldsymbol{x}_1) p(\boldsymbol{x}_1)\, d\boldsymbol{x}_1 \right) \\
&= \int (p_t(\boldsymbol{x}|\boldsymbol{x}_1) \nabla \cdot \boldsymbol{v}_t) p(\boldsymbol{x}_1)\, d\boldsymbol{x}_1 + \int (\boldsymbol{v}_t \cdot \nabla p_t(\boldsymbol{x}|\boldsymbol{x}_1)) p(\boldsymbol{x}_1)\, d\boldsymbol{x}_1 \\
&= \int (p_t(\boldsymbol{x}|\boldsymbol{x}_1) \nabla \cdot \boldsymbol{v}_t + \boldsymbol{v}_t \cdot \nabla p_t(\boldsymbol{x}|\boldsymbol{x}_1)) p(\boldsymbol{x}_1)\, d\boldsymbol{x}_1.
\end{aligned}
\tag{29}
$$

Combining equation (28) with equation (29), we deduce that:

$$
\begin{aligned}
&p_t\big(\nabla \cdot \boldsymbol{u}_t + \boldsymbol{u}_t \cdot \nabla \log p_t\big) - p_t\big(\nabla \cdot \boldsymbol{v}_t + \boldsymbol{v}_t \cdot \nabla \log p_t\big) \\
&= \left( \int \big(p_t(\boldsymbol{x}|\boldsymbol{x}_1) \nabla \cdot \boldsymbol{u}_t(\boldsymbol{x}|\boldsymbol{x}_1) + \boldsymbol{u}_t(\boldsymbol{x}|\boldsymbol{x}_1) \cdot \nabla p_t(\boldsymbol{x}|\boldsymbol{x}_1)\big) p(\boldsymbol{x}_1)\, d\boldsymbol{x}_1 \right) \\
&\quad - \left( \int \big(p_t(\boldsymbol{x}|\boldsymbol{x}_1) \nabla \cdot \boldsymbol{v}_t + \boldsymbol{v}_t \cdot \nabla p_t(\boldsymbol{x}|\boldsymbol{x}_1)\big) p(\boldsymbol{x}_1)\, d\boldsymbol{x}_1 \right) \\
&= \int (\nabla \cdot \boldsymbol{u}_t(\boldsymbol{x}|\boldsymbol{x}_1) - \nabla \cdot \boldsymbol{v}_t) p_t(\boldsymbol{x}|\boldsymbol{x}_1) p(\boldsymbol{x}_1)\, d\boldsymbol{x}_1 + \int \big(\boldsymbol{u}_t(\boldsymbol{x}|\boldsymbol{x}_1) - \boldsymbol{v}_t(\boldsymbol{x})\big) \cdot \nabla p_t(\boldsymbol{x}|\boldsymbol{x}_1) p(\boldsymbol{x}_1)\, d\boldsymbol{x}_1 \\
&= \int \left[ (\nabla \cdot \boldsymbol{u}_t(\boldsymbol{x}|\boldsymbol{x}_1) - \nabla \cdot \boldsymbol{v}_t) + \big(\boldsymbol{u}_t(\boldsymbol{x}|\boldsymbol{x}_1) - \boldsymbol{v}_t(\boldsymbol{x})\big) \cdot \nabla \log p_t(\boldsymbol{x}|\boldsymbol{x}_1) \right] p_t(\boldsymbol{x}|\boldsymbol{x}_1) p(\boldsymbol{x}_1)\, d\boldsymbol{x}_1.
\end{aligned}
\tag{30}
$$

Now from the definitions of $\mathcal{L}_{\mathrm{DM}}(\theta)$ and $\mathcal{L}_{\mathrm{CDM}}(\theta)$, we deduce that

$$
\begin{aligned}
\mathcal{L}_{\mathrm{DM}}(\theta) &= \int_0^T \int p_t \Big| \nabla \cdot (\boldsymbol{u}_t - \boldsymbol{v}_t) + (\boldsymbol{u}_t - \boldsymbol{v}_t) \cdot \nabla \log p_t \Big|\, d\boldsymbol{x}\, dt \\
&= \int_0^T \int \Big| p_t\big(\nabla \cdot \boldsymbol{u}_t + \boldsymbol{u}_t \cdot \nabla \log p_t\big) - p_t\big(\nabla \cdot \boldsymbol{v}_t + \boldsymbol{v}_t \cdot \nabla \log p_t\big) \Big|\, d\boldsymbol{x}\, dt \\
&= \int_0^T \int \left| \int \left[ (\nabla \cdot \boldsymbol{u}_t(\boldsymbol{x}|\boldsymbol{x}_1) - \nabla \cdot \boldsymbol{v}_t) + \big(\boldsymbol{u}_t(\boldsymbol{x}|\boldsymbol{x}_1) - \boldsymbol{v}_t(\boldsymbol{x})\big) \cdot \nabla \log p_t(\boldsymbol{x}|\boldsymbol{x}_1) \right] p_t(\boldsymbol{x}|\boldsymbol{x}_1) p(\boldsymbol{x}_1)\, d\boldsymbol{x}_1 \right|\, d\boldsymbol{x}\, dt \\
&\leq \int_0^T \int \int \left| (\nabla \cdot \boldsymbol{u}_t(\boldsymbol{x}|\boldsymbol{x}_1) - \nabla \cdot \boldsymbol{v}_t) + \big(\boldsymbol{u}_t(\boldsymbol{x}|\boldsymbol{x}_1) - \boldsymbol{v}_t(\boldsymbol{x})\big) \cdot \nabla \log p_t(\boldsymbol{x}|\boldsymbol{x}_1) \right| p_t(\boldsymbol{x}|\boldsymbol{x}_1) p(\boldsymbol{x}_1)\, d\boldsymbol{x}_1\, d\boldsymbol{x}\, dt \\
&= \mathcal{L}_{\mathrm{CDM}}(\theta).
\end{aligned}
\tag{31}
$$

$\square$

## B. Experiments Details

### B.1. Trajectory Sampling for Dynamical Systems

For this experiment, we repeatedly use the Dormand-Prince ODE solver with an absolute tolerance $1.4 \times 10^{-8}$ and relative tolerance $1 \times 10^{-6}$.

**Lorenz**  The Lorenz system (Lorenz, 1963) is a chaotic dynamical system given by

$$
\dot{\boldsymbol{x}} = \begin{bmatrix} \dot{x}_1 \\ \dot{x}_2 \\ \dot{x}_3 \end{bmatrix} = F(\boldsymbol{x}) = \begin{bmatrix} \sigma(x_2 - x_1) \\ x_1(\rho - x_3) - x_2 \\ x_1 x_2 - \beta x_3 \end{bmatrix}
$$

Following (Finzi et al., 2023), we set $\sigma = 10$, $\rho = 28$ and $\beta = 8/3$, and we used a scaled version of the Lorenz system to bound the system components $x_i$ to $[-3, 3]$ for $i \in \{1, 2, 3\}$ while preserving the original dynamics. The scaled system is given by $\tilde{F}(\boldsymbol{x}) = F(20\boldsymbol{x})/20$.

**FitzHugh-Nagumo**    The FitzHugh-Nagumo system (FitzHugh, 1961; Nagumo et al., 1962) is a dynamical system modeling an excitable neuron and is given by

$$\dot{x}_i = x_i(a_i - x_i)(x_i - 1) - y_i + k\sum_{j=1}^{d} A_{ij}(x_j - x_i)$$

$$\dot{y}_i = b_i x_i - c_i y_i$$

for $i \in \{1, 2\}$. Following (Farazmand & Sapsis, 2019; Finzi et al., 2023), the parameters are set as follows: $a_1 = a_2 = -0.025794$, $b_1 = 0.0065$, $b_2 = 0.0135$, $c_1 = c_2 = 0.2$, $k = 0.128$, and $A_{ij} = 1 - \delta_{ij}$ where $\delta$ is the Kronecker delta.

**Trajectory Dataset Construction**    Trajectories for the dataset are computed using the ODE solver. The trajectories' initial conditions are sampled from Gaussian distributions – $\mathcal{N}(\mathbf{0}, \mathbf{I})$ for Lorenz, and $\mathcal{N}(\mathbf{0}, (0.2)^2\mathbf{I})$ for FitzHugh-Nagumo. Each trajectory has 60 consecutive and evenly spaced time steps, where the first time step occurs after some trajectory "burn-in" time to allow the system to reach its stationary trajectory distribution. The first 30 and 250 time steps computed by the ODE solver are "burn-in" for Lorenz and FitzHugh-Nagumo, respectively. The time step sizes are 0.1 and 6.0, respectively.

**Model Hyperparameters and Training**    All the models used the same UNet architecture as in (Finzi et al., 2023), and we used a variance exploding schedule (Song et al., 2020). We train the models on a training set of 32,000 trajectories computed by the ODE solver using Adam for 2,000 epochs with a batch size of 500. For FM and FDM, we also used an exponential decay learning rate scheduler with a decay rate of 0.995. The initial learning rate for the diffusion model and FM was $10^{-4}$. The learning rate and regularization coefficients for FDM were tuned using Optuna (Akiba et al., 2019) for the lowest CFM loss produced by the EMA parameters and are given in Table 10. We sampled the times for the diffusion and CFM loss on a shifted grid following (Finzi et al., 2023).

| Dynamical system | Learning rate | $\lambda_1$ | $\lambda_2$ |
|---|---|---|---|
| Lorenz | 0.000796 | 1 | 0.000385 |
| FitzHugh-Nagumo | 0.000245 | 1 | 0.00552 |

*Table 10.* Learning rate and regularization coefficient used to train FDM for Lorenz and FitzHugh-Nagumo dynamical systems.

We evaluated the models with the exponential moving average (EMA) of the parameters with a 2,000 epoch period.

**Loss Weighting Functions**    The loss of the diffusion model is equation (7) of (Song et al., 2020) where we used $\lambda(t) = \sigma_t^2$ as the weighting. For both FM and FDM, the term $\mathcal{L}_{\text{CFM}}$ in their loss was weighted by $1/(\sigma'_{1-t})^2$. The term $\mathcal{L}_{\text{CDM}}$ in the loss of FDM was weighted by $\sigma_{1-t}/(\sigma'_{1-t}Md)$ where $M = 60$ is the number of trajectory time steps and $d$ is the dimension of the dynamical system.

**Estimating the Divergence**    We estimated the divergence of FDM with respect to its trajectory input using the Hutchinson tracer estimator (Hutchinson, 1989; Grathwohl et al., 2018) where the noise vector is sampled from $\mathcal{N}(\mathbf{0}, \mathbf{I})$.

**Likelihood Estimation**    We computed a test set of 32,000 trajectories using the ODE solver and evaluated their log-likelihood using the continuous change-of-variables formula from (Grathwohl et al., 2018) with the ODE solver. Table 6 was produced by computing the mean log-likelihood over the trajectories and their dimension.

### B.2. DNA Sequence Generation

In this task, the model approximates a classifier

$$\hat{p}(\boldsymbol{x}_1|\boldsymbol{x}, \theta) \approx \frac{p_t(\boldsymbol{x}|\boldsymbol{x}_1)p(\boldsymbol{x}_1)}{p_t(\boldsymbol{x})} \tag{32}$$

instead of directly approximating the vector field $\hat{\boldsymbol{v}}_t(\boldsymbol{x}, \theta) \approx \boldsymbol{u}_t(\boldsymbol{x})$. Then, it constructs a vector field based on the classifier as follows:

$$\hat{\boldsymbol{v}}_t(\boldsymbol{x}, \theta) = \sum_{i=1}^{K} \boldsymbol{u}_t(\boldsymbol{x}|\boldsymbol{x}_1 = \boldsymbol{e}_i)\hat{p}(\boldsymbol{x}_1 = \boldsymbol{e}_i|\boldsymbol{x}, \theta), \tag{33}$$

where $K$ is the number of categories and the divergence term is given by

$$\nabla_{\boldsymbol{x}} \cdot \hat{\boldsymbol{v}}_t(\boldsymbol{x}, \theta) = \sum_{i=1}^{K} \left[ \left\langle \nabla p(\boldsymbol{x}_1 = \boldsymbol{e}_i | \boldsymbol{x}, \theta), \boldsymbol{u}_t(\boldsymbol{x} | \boldsymbol{x}_1 = \boldsymbol{e}_i) \right\rangle + p(\boldsymbol{x}_1 = \boldsymbol{e}_i | \boldsymbol{x}, \theta) \nabla_{\boldsymbol{x}} \cdot \boldsymbol{u}_t(\boldsymbol{x} | \boldsymbol{x}_1 = \boldsymbol{e}_i) \right] \tag{34}$$

If we directly learn $\nabla_{\boldsymbol{x}} \cdot \hat{\boldsymbol{v}}_t(\boldsymbol{x}, \theta)$, it requires computing $\nabla_{\boldsymbol{x}} \cdot \left[ \boldsymbol{u}_t(\boldsymbol{x} | \boldsymbol{x}_1 = \boldsymbol{e}_i) \hat{p}(\boldsymbol{x}_1 = \boldsymbol{e}_i | \boldsymbol{x}, \theta) \right]$ for $i = 1, 2, ..., K$ which can be very expensive in memory footprint and time consumption. Furthermore, notice that $\boldsymbol{u}_t(\boldsymbol{x} | \boldsymbol{x}_1 = \boldsymbol{e}_i)$ is a pre-defined vector field that is independent of parameters $\theta$ and so is $\nabla_{\boldsymbol{x}} \cdot \boldsymbol{u}_t(\boldsymbol{x} | \boldsymbol{x}_1 = \boldsymbol{e}_i)$. Thus, there is no need to learn it.

For $p(\boldsymbol{x}_1 = \boldsymbol{e}_i | \boldsymbol{x}, \theta)$, Appendix A of (Stark et al., 2024) states that $\hat{\boldsymbol{v}}_t(\boldsymbol{x}, \theta)$ approximates the vector field if $\hat{p}(\boldsymbol{x}_1 | \boldsymbol{x}, \theta)$ ideally approximates the classifier $p(\boldsymbol{x}_1 | \boldsymbol{x})$. Consider an ideal classifier $p(\boldsymbol{x}_1 = \boldsymbol{e}_i | \boldsymbol{x})$ for class $\boldsymbol{x}_1 = \boldsymbol{e}_i$, then $p(\boldsymbol{x}_1 = \boldsymbol{e}_i | \boldsymbol{x}) = 1$ if $\boldsymbol{x}$ belongs to class $\boldsymbol{x}_1$ else 0. Let $\boldsymbol{x} \in D$, where $D$ is the domain of this classifier, then we have

- $p(\boldsymbol{x}_1 = \boldsymbol{e}_i | \boldsymbol{x})$ is not continuous in $D$.

- Suppose $D_1$ is the union of all the differentiable sub-domains of $D$, then $\nabla_{\boldsymbol{x}} p(\boldsymbol{x}_1 = \boldsymbol{e}_i | \boldsymbol{x}) = \boldsymbol{0}$ for $\boldsymbol{x} \in D_1$.

Therefore, the remaining thing is to include $\|\nabla_{\boldsymbol{x}} p(\boldsymbol{x}_1 = \boldsymbol{e}_i | \boldsymbol{x})\|$ for $\boldsymbol{x} \in D_1$ in the training objective. In practice, we train the classifier by empirically estimating the cross entropy based on the perturbed points $\boldsymbol{x}$ with its corresponding initial data $\boldsymbol{x}_1$ as the class label. We can just assume any point in a sufficiently small ball around such a perturbed data point $\boldsymbol{x}$ belongs to the same class $\boldsymbol{x}_1$ so the classifier is differentiable inside this ball, then we penalize $\|\nabla_{\boldsymbol{x}} \hat{p}(\boldsymbol{x}_1 | \boldsymbol{x}, \theta)\|$ in training the model.

**Promoter Data** We use a dataset of 100,000 promoter sequences with 1,024 base pairs extracted from a database of human promoters (Hon et al., 2017). Each sequence has a CAGE signal (Shiraki et al., 2003) annotation available from the FANTOM5 promoter atlas, which indicates the likelihood of transcription initiation at each base pair. Sequences from chromosomes 8 and 9 are used as a test set, and the rest for training.

**Model Hyperparameters and Training** We just follow the experimental setup of (Stark et al., 2024). For the simplex dimension toy experiment, we train all models for 450,000 steps with a batch size of 512 to ensure that they have all converged and then evaluate the KL of the final step. For promoter design, we train for 200 epochs with a learning rate of $5 \times 10^{-4}$ and early stopping on the MSE on the validation set. We use 100 inference steps for generation. Table 11 show how we set $\lambda_1$ and $\lambda_2$ for divergence loss.

| Tasks | Learning rate | $\lambda_1$ | $\lambda_2$ |
|---|---|---|---|
| Simplex Dimension | $5 \times 10^{-4}$ | 0.5 | 0.05 |
| Promoter Design | $5 \times 10^{-4}$ | 1 | 0.01 |

*Table 11.* Learning rate and regularization coefficient used to train FDM for DNA sequence.

### B.3. Generative Modeling for Videos

We follow the experimental setting and models used in (Davtyan et al., 2023).

**Architechture** We use U-ViT (Bao et al., 2023) to model the flow matching vector field and use VQGAN (Esser et al., 2021) to encode (resp. decode) each frame of the video to (resp. from) the latent space with the following configurations

**Model Hyperparameters** See Table 13.

## C. Additional numerical results

### C.1. Dirichlet Flow Matching

Table 14 shows the test KL divergence of models for the simplex dimension toy experiment of DNA sequence generation.

| Parameter | KTH | BAIR |
|---|---|---|
| embed_dim | 4 | 4 |
| n_embed | 16384 | 16384 |
| double_z | False | False |
| z_channels | 4 | 4 |
| resolution | 64 | 64 |
| in_channels | 3 | 3 |
| out_ch | 3 | 3 |
| ch | 128 | 128 |
| ch_mult | [1,2,2,4] | [1,2,2,4] |
| num_res_blocks | 2 | 2 |
| attn_resolutions | [16] | [16] |
| dropout | 0.0 | 0.0 |
| disc_conditional | False | False |
| disc_in_channels | 3 | 3 |
| disc_start | 20k | 20k |
| disc_weight | 0.8 | 0.8 |
| codebook_weight | 1.0 | 1.0 |

*Table 12.* Parameters of VQGAN for the KTH dataset.

| Hyperparameter | Values/Search Space |
|---|---|
| Iterations | 300000 |
| Batch size | [16, 32, 64] |
| Learning rate | [2e-4, 2e-5] |
| Learning rate scheduler | polynomial |
| Learning rate decay power | 0.5 |
| Weight decay rate | 1e-12 |
| $\lambda_1$, $\lambda_2$ | [[0.5, 1e-2], [1, 1e-2]] |

*Table 13.* Training hyperparameters of video prediction.

| Method | KL Divergence |
|---|---|
| Linear FM | $2.5_{\pm 0.1}$E-2 |
| Linear FDM | $2.1_{\pm 0.1}$E-2 |
| Dirichlet FM | $1.8_{\pm 0.1}$E-2 |
| Dirichlet FDM | $1.5_{\pm 0.1}$E-2 |

*Table 14.* KL divergence of the generated distribution to the target distribution.

### C.2. Flow Matching for User-defined Events

**Unguided Sampling Histograms:** The histograms of the event constraint values for the trajectories sampled without guidance by each model are shown in Fig. 7.

**KL Divergence** Table 7 shows the KL divergence between the histogram distributions. For Lorenz, FDM's unguided sampling has the lowest KL divergence, with the divergence of the diffusion model and FM being 0.0007 and 0.0032 larger. In guided sampling, the FM has a lower KL divergence than the diffusion model and FDM by about 0.02 and 0.05, respectively. For FitzHugh-Nagumo, the diffusion model has a lower KL divergence than FM and FDM by 0.002. In guided sampling, FDM attains the largest performance gap with a KL divergence of about 0.1 and 0.14 lower than the diffusion model and FM, respectively.

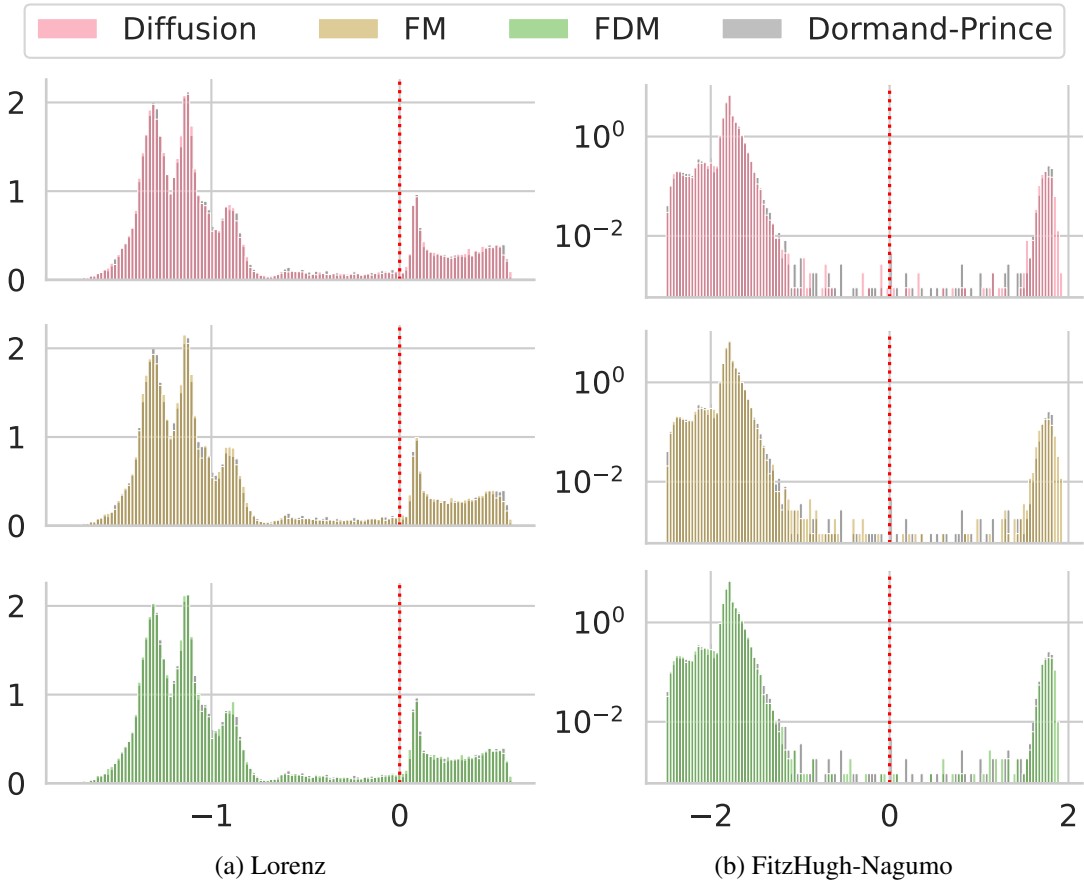

(a) Lorenz  (b) FitzHugh-Nagumo

*Figure 7.* Histograms of the event constraint $C$ evaluated on the data and trajectories generated from the models.

| | Lorenz | | FitzHugh-Nagumo | |
|---|---|---|---|---|
| Model | $p(\boldsymbol{x}_1)$ | $p(\boldsymbol{x}_1\|E)$ | $p(\boldsymbol{x}_1)$ | $p(\boldsymbol{x}_1\|E)$ |
| Diffusion | 0.0056 | 0.2774 | 0.0260 | 0.3011 |
| FM | 0.0081 | 0.2560 | 0.0280 | 0.3468 |
| FDM | 0.0049 | 0.3045 | 0.0280 | 0.2084 |

*Table 15.* KL divergence between the histograms of the event constraint value $C(\boldsymbol{x}_1)$ for event trajectories $\boldsymbol{x}_1$ in the dataset of trajectories computed by an ODE solver and event trajectories sampled with event guidance from the models.

## D. Efficient Squared Loss

We observe that the conditional divergence loss in equation (14) is an absolute-value objective, whose non-differentiability at the origin removes smoothness and can be less efficient than squared-error alternatives. In practice, we replace it with the following squared loss:

$$\mathcal{L}_{\text{CDM-2}}(\theta) = \mathbb{E}_{t,p_t(\boldsymbol{x}|\boldsymbol{x}_1),p(\boldsymbol{x}_1)}\left[\left|\left(\nabla \cdot \boldsymbol{u}_t(\boldsymbol{x}|\boldsymbol{x}_1) - \nabla \cdot \boldsymbol{v}_t(\boldsymbol{x},\theta)\right) + \left(\boldsymbol{u}_t(\boldsymbol{x}|\boldsymbol{x}_1) - \boldsymbol{v}_t(\boldsymbol{x},\theta)\right) \cdot \nabla \log p_t(\boldsymbol{x}|\boldsymbol{x}_1)\right|^2\right], \quad (35)$$

**Theorem D.1** (Upper Bound for $\mathcal{L}_{\text{CDM}}$). *The squared loss $\mathcal{L}_{\text{CDM-2}}(\theta)$ provides an upper bound:*

$$\mathcal{L}_{\text{CDM}}(\theta) \leq \sqrt{\mathcal{L}_{\text{CDM-2}}(\theta)} \quad (36)$$

*Proof.* Let

$$f(\boldsymbol{x},\boldsymbol{x}_1,t) = \left(\nabla \cdot \boldsymbol{u}_t(\boldsymbol{x}|\boldsymbol{x}_1) - \nabla \cdot \boldsymbol{v}_t(\boldsymbol{x},\theta)\right) + \left(\boldsymbol{u}_t(\boldsymbol{x}|\boldsymbol{x}_1) - \boldsymbol{v}_t(\boldsymbol{x},\theta)\right) \cdot \nabla \log p_t(\boldsymbol{x}|\boldsymbol{x}_1)$$

we have

$$
\begin{aligned}
\mathcal{L}_{\text{CDM}}(\theta) &= \int \int \int |f(\boldsymbol{x}, \boldsymbol{x}_1, t)| p_t(\boldsymbol{x}|\boldsymbol{x}_1) p(\boldsymbol{x}_1) d\boldsymbol{x} d\boldsymbol{x}_1 dt \\
&= \int \int \int |f(\boldsymbol{x}, \boldsymbol{x}_1, t)| p(\boldsymbol{x}|\boldsymbol{x}_1, t) p(\boldsymbol{x}_1) p(t) d\boldsymbol{x} d\boldsymbol{x}_1 dt \\
&\underbrace{\leq}_{\text{Cauchy-Schwarz Ineq.}} \left( \int \int \int f^2(\boldsymbol{x}, \boldsymbol{x}_1, t) p(\boldsymbol{x}|\boldsymbol{x}_1, t) p(\boldsymbol{x}) p(t) d\boldsymbol{x} d\boldsymbol{x}_1 dt \right)^{\frac{1}{2}} \\
&= \sqrt{\mathcal{L}_{\text{CDM-2}}(\theta)}
\end{aligned}
\tag{37}
$$

where we define $p_t(\boldsymbol{x}|\boldsymbol{x}_1) := p(\boldsymbol{x}|\boldsymbol{x}_1, t) p(t)$. $\qquad\square$

### D.1. Efficient Squared Loss for High-dimensional Data

#### D.1.1. ESTIMATED OBJECTIVE VIA HUTCHINSON TRACE ESTIMATOR

Let $d$ be the data dimension. The second-order term in equation (35) requires computing the trace of the full Jacobian of the vector regressor model $\boldsymbol{v}_t(\boldsymbol{x}, \theta)$, which typically incurs a computational time complexity of $\mathcal{O}(d^2)$ and becomes impractical for high-dimensional data. Following (Lu et al., 2022; Lai et al., 2023), this cost can be reduced to $\mathcal{O}(d)$ by employing Hutchinson's trace estimator (Hutchinson, 1989) and automatic differentiation (Paszke et al., 2017) provided by general deep learning frameworks, requiring only a single backpropagation pass.

For a $d$-by-$d$ matrix $\boldsymbol{A}$, its trace can be unbiasedly estimated by

$$
\text{tr}(\boldsymbol{A}) = \mathbb{E}_{p(\boldsymbol{\varepsilon})}\big[\boldsymbol{\varepsilon}^\top \boldsymbol{A} \boldsymbol{\varepsilon}\big] = \mathbb{E}_{p(\boldsymbol{\varepsilon})}\big[\boldsymbol{\varepsilon} \cdot (\boldsymbol{A} \cdot \boldsymbol{\varepsilon})\big]
$$

where $p(\boldsymbol{\varepsilon})$ is a $d$-dimensional standard Gaussian. Then we proposed the following estimation:

$$
\begin{aligned}
\mathcal{L}_{\text{CDM-2}}^{\text{est}}(\theta) = \mathbb{E}_{t, p_t(\boldsymbol{x}|\boldsymbol{x}_1), p(\boldsymbol{x}_1), p(\boldsymbol{\varepsilon})}\Big[\Big| \boldsymbol{\varepsilon} \cdot \big(\nabla \boldsymbol{u}_t(\boldsymbol{x}|\boldsymbol{x}_1) \cdot \boldsymbol{\varepsilon} - \nabla \boldsymbol{v}_t(\boldsymbol{x}, \theta) \cdot \boldsymbol{\varepsilon}\big) \\
+ \big(\boldsymbol{u}_t(\boldsymbol{x}|\boldsymbol{x}_1) \cdot \boldsymbol{\varepsilon} - \boldsymbol{v}_t(\boldsymbol{x}, \theta) \cdot \boldsymbol{\varepsilon}\big)\big(\nabla \log p_t(\boldsymbol{x}|\boldsymbol{x}_1) \cdot \boldsymbol{\varepsilon}\big)\Big|^2\Big],
\end{aligned}
\tag{38}
$$

where $\nabla \boldsymbol{u}_t(\boldsymbol{x}|\boldsymbol{x}_1)$ and $\nabla \log p_t(\boldsymbol{x}|\boldsymbol{x}_1)$ are already given pre-defined matrix and vector. The term $\nabla \boldsymbol{v}_t(\boldsymbol{x}, \theta) \cdot \boldsymbol{\varepsilon}$ can be efficiently computed by the `jvp` interface, such as `torch.func.jvp` in PyTorch or `jax.jvp` in JAX.

**Theorem D.2** (Upper Bound for $\mathcal{L}_{\text{CDM}-2}$). *The estimated squared loss $\mathcal{L}_{\text{CDM-2}}^{\text{est}}(\theta)$ provides an upper bound:*

$$
\mathcal{L}_{\text{CDM-2}}(\theta) \leq \mathcal{L}_{\text{CDM-2}}^{\text{est}}(\theta)
\tag{39}
$$

*Proof.*

$$
\begin{aligned}
\mathcal{L}_{\text{CDM-2}}(\theta) &= \mathbb{E}_{t, p_t(\boldsymbol{x}|\boldsymbol{x}_1), p(\boldsymbol{x}_1)}\Big[\Big| \big(\nabla \cdot \boldsymbol{u}_t(\boldsymbol{x}|\boldsymbol{x}_1) - \nabla \cdot \boldsymbol{v}_t(\boldsymbol{x}, \theta)\big) + \big(\boldsymbol{u}_t(\boldsymbol{x}|\boldsymbol{x}_1) - \boldsymbol{v}_t(\boldsymbol{x}, \theta)\big) \cdot \nabla \log p_t(\boldsymbol{x}|\boldsymbol{x}_1) \Big|^2\Big] \\
&= \mathbb{E}_{t, p_t(\boldsymbol{x}|\boldsymbol{x}_1), p(\boldsymbol{x}_1)}\Big[\Big| \mathbb{E}_{p(\boldsymbol{\varepsilon})}\Big[\boldsymbol{\varepsilon}^\top \big(\nabla \boldsymbol{u}_t(\boldsymbol{x}|\boldsymbol{x}_1) - \nabla \boldsymbol{v}_t(\boldsymbol{x}, \theta)\big)\boldsymbol{\varepsilon} \\
&\qquad\qquad\qquad\qquad\qquad + \boldsymbol{\varepsilon}^\top \big(\big(\boldsymbol{u}_t(\boldsymbol{x}|\boldsymbol{x}_1) - \boldsymbol{v}_t(\boldsymbol{x}, \theta)\big) \otimes \nabla \log p_t(\boldsymbol{x}|\boldsymbol{x}_1)\big)\boldsymbol{\varepsilon}\Big]\Big|^2\Big] \\
&\underbrace{\leq}_{\text{Cauchy-Schwarz Ineq.}} \mathbb{E}_{t, p_t(\boldsymbol{x}|\boldsymbol{x}_1), p(\boldsymbol{x}_1), p(\boldsymbol{\varepsilon})}\Big[\Big| \boldsymbol{\varepsilon}^\top \big(\nabla \boldsymbol{u}_t(\boldsymbol{x}|\boldsymbol{x}_1) - \nabla \boldsymbol{v}_t(\boldsymbol{x}, \theta)\big)\boldsymbol{\varepsilon} \\
&\qquad\qquad\qquad\qquad\qquad + \boldsymbol{\varepsilon}^\top \big(\big(\boldsymbol{u}_t(\boldsymbol{x}|\boldsymbol{x}_1) - \boldsymbol{v}_t(\boldsymbol{x}, \theta)\big) \otimes \nabla \log p_t(\boldsymbol{x}|\boldsymbol{x}_1)\big)\boldsymbol{\varepsilon}\Big|^2\Big] \\
&= \mathbb{E}_{t, p_t(\boldsymbol{x}|\boldsymbol{x}_1), p(\boldsymbol{x}_1), p(\boldsymbol{\varepsilon})}\Big[\Big| \boldsymbol{\varepsilon} \cdot \big(\nabla \boldsymbol{u}_t(\boldsymbol{x}|\boldsymbol{x}_1) \cdot \boldsymbol{\varepsilon} - \nabla \boldsymbol{v}_t(\boldsymbol{x}, \theta) \cdot \boldsymbol{\varepsilon}\big) \\
&\qquad\qquad\qquad\qquad\qquad + \big(\boldsymbol{u}_t(\boldsymbol{x}|\boldsymbol{x}_1) \cdot \boldsymbol{\varepsilon} - \boldsymbol{v}_t(\boldsymbol{x}, \theta) \cdot \boldsymbol{\varepsilon}\big)\big(\nabla \log p_t(\boldsymbol{x}|\boldsymbol{x}_1) \cdot \boldsymbol{\varepsilon}\big)\Big|^2\Big] \\
&= \mathcal{L}_{\text{CDM-2}}^{\text{est}}(\theta)
\end{aligned}
\tag{40}
$$

$\qquad\square$

### D.1.2. STOP GRADIENT

In practice, a stop-gradient operation is applied on the $\boldsymbol{v}_t(\boldsymbol{x}, \theta)$ in $\mathcal{L}_{\text{CDM-2}}^{\text{est}}(\theta)$ following common practice (Frans et al., 2024; Lu et al., 2022; Song & Dhariwal, 2023). In our case, we train the model by combining $\mathcal{L}_{\text{CFM}}(\theta)$ and the conditional divergence matching loss as discussed in Section 4 so the stop-gradient operation eliminates the need for "double backpropagation" through $\boldsymbol{v}_t(\boldsymbol{x}, \theta)$, making the training more efficient. So we define the squared efficient conditional divergence matching loss:

$$
\mathcal{L}_{\text{CDM-2}}^{\text{eff}}(\theta) = \mathbb{E}_{t, p_t(\boldsymbol{x}|\boldsymbol{x}_1), p(\boldsymbol{x}_1), p(\boldsymbol{\varepsilon})} \Big[ \Big| \boldsymbol{\varepsilon} \cdot \big( \nabla \boldsymbol{u}_t(\boldsymbol{x}|\boldsymbol{x}_1) \cdot \boldsymbol{\varepsilon} - \nabla \boldsymbol{v}_t(\boldsymbol{x}, \theta) \cdot \boldsymbol{\varepsilon} \big)
$$
$$
+ \big( \boldsymbol{u}_t(\boldsymbol{x}|\boldsymbol{x}_1) \cdot \boldsymbol{\varepsilon} - \text{sg}\big( \boldsymbol{v}_t(\boldsymbol{x}, \theta) \big) \cdot \boldsymbol{\varepsilon} \big) \big( \nabla \log p_t(\boldsymbol{x}|\boldsymbol{x}_1) \cdot \boldsymbol{\varepsilon} \big) \Big|^2 \Big], \tag{41}
$$

where $\text{sg}$ denotes stop-gradient operator, which prevents gradients from propagating to $\theta$ through the term $\boldsymbol{v}_t(\boldsymbol{x}, \theta)$ in $\mathcal{L}_{\text{CDM-2}}^{\text{eff}}(\theta)$. Thus, optimizing the flow and divergence matching loss in equation (17) requires only one extra backward pass compared to the baseline $\mathcal{L}_{\text{CFM}}$.

