# OpenReview forum: "Improving Flow Matching by Aligning Flow Divergence"
_ICML.cc/2025/Conference — ICML 2025 poster_

### Official Review · Reviewer_WCvg · 2025-03-12

**Overall Recommendation:** 4

**Summary:**

The paper makes a very keen insight that the "TRUE" goal of flow matching is to
approximate the probabilty time series $(t \mapsto p_t)$ with the approximate probability time series
$(t \mapsto \hat{p}_t)$.  In doing so, they begin from the  difference
$\hat{p}_t - p_t$  and that, in order to make this small in terms of total variation, one must not only
align the corresponding vector field v that satisfies the CE wrt  p_t, but also the divergence.
In particular if $(p_t, v_t)$ and  $(\hat{p}_t, \hat{v}_t)$ are both the CE satisfying solutions, then both
- $| v- \hat v|$ as wellas
- $| \nabla \cdot v- \nabla \cdot \hat{v} |$
must become small.

Just as there is a conditional-localized counterpart ($L_{CFM}$) for  the loss of $|v- \hat v|$  ($L_{FM}$)through the trick resembling Denoising score matching,  they construct the localized counterpart $L_{CDM}$ for the original target about $| \nabla \cdot v- \nabla \cdot \hat{v} |$ ($L_{DM}$) .
Their proposal is to balance both of them and simultaneoulsy optimize them at the same time.
Unlike $L_{CFM}$ that is known to be just a scaler different from $L_{FM}$,  they were able to show only $L_{DM} < L_{CDM}$.
But this inequality is in the friendly direction, because LHS can be made smaller by making RHS smaller.


They first show that this is obviously effective on synthetic dataset in terms of both final accuracy and the speed of learning.
They showcase their result on categorical domain (DNA-Sequence) as well as PDE, and even Video Prediction with latent modeling.

**Claims And Evidence:**

Their claims are mathematically proven. The efficacy of the model is proven both on synthetic dataset,
dataset of continuous domain, dataset of discrete domain, and dataset of practically an infinite dimensional domain.

**Essential References Not Discussed:**

Nothing in particular that came to my mind.

**Experimental Designs Or Analyses:**

Experiment on the Dynamical system and Video prediction seems particulaly well designed and their evaluations seems to be in alignment with the conventional metric.

**Methods And Evaluation Criteria:**

The proposed method is general in nature, and I believe that it shall be evaluated not on the basis of the SOTA result on each domain but on the basis that it can make improvement on the competitive method based on FM, which is done properly with reasonable metric.

**Other Comments Or Suggestions:**

Please see the section above.

**Other Strengths And Weaknesses:**

## Strengths:
- The paper is well motivated
- The proposition is convincing, and it is theoretically supported.
- When the exact same technique as CFM did not work for the DM, the paper proposed a work-around solution of upper bound, which is very noble
-  experiments are well designed, and the results are promising

## Weakness:
-  I will ask a question or two in the questions section
-  While the paper's proposition is convincing, the proposition introduces an hyperparameter(the balancing of CDM and CFM)  whose meaning is a little hard to interpret in terms of data (it is clear in terms of definition, but its choice seems hard to make based on the nature of the data itself).    This might however be somehow inferrable from the choice of the path distributions from which to create the supervisory signal (using OT, for example, make a difference about the optimal choice of $\lambda$.  It will be great if some knowledge can be shared regarding the relation between the choice of conditional vector field and the suggested choice of $\lambda$s.

**Questions For Authors:**

### Q1
While I find the localization of DM with CDM very inspiring, as is noted in the paper this requires the computation of $p(x|x_1)$, which in the pure FM setting is a delta function, which would cause a problem. Indeed, this is not an essential problem of the philosophy of the work itself, beause DM only looks at $\nabla \log p_t(x)$,  in which $p_t$ is guaranteed to be absolutely continuous whenever $p_0$ is.

As is mentioned in the early part of the manuscript,  the paper is resolving this problem by assuming the path to be taking the form of
$$ \psi(t | x_1, x_source, \epsilon) = (1-t) x_{source}  + t x_1 +  \sigma_t \epsilon$$
where $\epsilon$ is a Gauss, and in experiments $\sigma_t$ is modeled in VP or VE way, and derive the corresponding vector field.
But this $\sigma_t$ looks a bit artificial, especially when $x_{source}$ is not Gaussian, for example--- and one of the strength of FM
lies in its application of mapping an arbitrary distribution to another.

It feels that, when the method is combined with the works like (Simulation-free Schrödinger bridges via score and flow matching), things would also align really well.  Has this work been considered to be applied to Schrodinger Bridge?

### Q2
Howq does the computational overhead scale by the addition of $L_{CDM}$? I very much believe that the philosophy of this work deserves much attention, but the gradient of $\nabla \cdot \hat{v | \theta}$ would, in the process of backpropagation, require second derivatives, and that sounds a bit scary from the engineering perspectives when we think of scaling this up to the level of very large networks. Would you please comment on that?

**Relation To Broader Scientific Literature:**

NA

**Theoretical Claims:**

Theoretical claims seems mostly valid on the appendix.

---

> ### Author Rebuttal · Authors · 2025-03-30
>
> Thank you for your thoughtful review and valuable feedback. We have revised the paper according to all reviewers’ feedback. In what follows, we provide point-by-point responses to your comments.
>
> ----
>
> **Q1. It will be great if some knowledge can be shared regarding the relation between the choice of conditional vector field and the suggested choice of $\lambda$s.**
>
> **Response:** It is an interesting comment. Our design of the new loss function - Equation (19) of our submission - is based on the empirical observation and discussion in Lines 268-274 (left column) and 220-229 (right column) of our submission. During our submission, we did not consider the meaning of $\lambda$s in terms of data, but it is an interesting future direction to design a principle to choose $\lambda$s optimally. We have acknowledged this point in the revised manuscript.
>
> ----
>
> **Q2. While I find the localization of DM with CDM very inspiring, … It feels that, when the method is combined with the works like (Simulation-free Schrodinger bridges via score and flow matching), things would also align really well. Has this work been considered to be applied to Schrodinger Bridge?**
>
> **Response:** Thank you for the very insightful comments. We have not considered the Schrodinger bridge yet, while we believe it is indeed an interesting research problem. We have briefly discussed this and cited the related reference that the reviewer pointed out in the revised manuscript.
>
> In the flow matching setting, the conditional probability path and the associated vector field are often designed on a per-sample basis. Therefore, our proposed CDM is tractable, similar to the baseline conditional flow matching.
>
>
> ----
>
> **Q3. How does the computational overhead scale by the addition of $L_{CDM}$? The gradient of $\nabla\cdot \hat v(x,\theta)$ would, in the process of backpropagation, require second derivatives, and that sounds a bit scary from the engineering perspectives when we think of scaling this up to the level of very large networks. Would you please comment on that?**
>
> **Response:** In practice, the divergence term $\nabla\cdot \hat v(x,\theta)$ is computed efficiently using the Hutchinson estimator (see Lines 307-309 in our submission).  We follow the approach used in [1] and [2], which employs the Hutchinson estimator to approximate the divergence term. Then, we only need to compute the derivative of a scalar-valued output with respect to the input to construct the loss, which involves a single call to torch.autograd.grad() and adds just one additional backward pass. Since we're only doing one extra backward pass, the additional memory footprint and computational time of computing the divergence loss scales similarly to that of the original CFM loss with respect to the data dimensionality. We compared the training times and peak memory usage, and found that incorporating the divergence loss does not significantly hinder the model's usability, with computational time and memory footprint remaining mostly within 1.5 times that of the original CFM.
>
> [1] Lu, Cheng, et al. "Maximum likelihood training for score-based diffusion odes by high order denoising score matching." International Conference on Machine Learning. ICML, 2022. https://arxiv.org/pdf/2206.08265
> [2] Lai, Chieh-Hsin, et al. "Fp-diffusion: Improving score-based diffusion models by enforcing the underlying score fokker-planck equation." International Conference on Machine Learning. ICML, 2023. https://arxiv.org/pdf/2210.04296
>
> ----
>
> Thank you for considering our rebuttal.

---

### Official Review · Reviewer_rKjb · 2025-03-13

**Overall Recommendation:** 3

**Summary:**

The paper proposes a very simple KL loss combined with CFM loss to improve the training of flow-based models. The optimizing results are very general across different tasks with a basic improvement.

**Claims And Evidence:**

NA

**Essential References Not Discussed:**

NA

**Experimental Designs Or Analyses:**

NA

**Methods And Evaluation Criteria:**

method

**Other Comments Or Suggestions:**

1\ The organization of the contribution part should be improved.
2\ Quantify the individual impact of the divergence loss term L_cdm on performance.

**Other Strengths And Weaknesses:**

S:
1\ Demonstrates consistent improvements over CFM across diverse tasks (synthetic data, dynamical systems, DNA, videos) with minimal computational overhead.
2\ Avoids costly higher-order score matching, making divergence alignment computationally practical.
W:
1\ Lack of implementation details which may indicate unfair comparisons.

**Questions For Authors:**

1\ How does FDM’s training time and memory footprint scale with data dimensionality especially for high-resolution videos?
2\ Can FDM be applied to non-Gaussian or non-OT probability paths

**Relation To Broader Scientific Literature:**

NA

**Theoretical Claims:**

NA

---

> ### Author Rebuttal · Authors · 2025-03-30
>
> Thank you for your thoughtful review and valuable feedback. We have revised the paper according to all reviewers’ feedback. In what follows, we provide point-by-point responses to your comments.
>
> ----
>
> **Q1. Lack of implementation details which may indicate unfair comparisons.**
>
> **Response:** In our submission, we have provided experimental and implementation details in Appendix B. To ensure fair comparisons with benchmarks, we use the same neural network architectures and model hyperparameters as the baseline methods. In particular, we use the same dataset and splitting, and employ the same search space for the batch size with the same training iterations.
>
> ---
>
> **Q2. The organization of the contribution part should be improved.**
>
> **Response:** Thank you for your suggestion. We have made the contribution section more concise and bulleted the key points in the revised paper.
>
> ---
>
> **Q3. Quantify the individual impact of the divergence loss term $L_{CDM}$ on performance.**
>
> **Response:** Our training loss is a weighted sum of $L_{CFM}$ and $L_{CDM}$ as shown in Equation (19) of our submission. Only using $L_{CFM}$ is the same as classic conditional flow matching, while, as we state in Lines 268-270 of our manuscript, directly minimizing $L_{CDM}$ cannot yield appealing results. Detailed explanations are given after those lines. In particular, we notice empirically that training the model by minimizing $L_{CDM}$ alone can be quite noisy.
>
> Using a weighted sum of $L_{CFM}$ and $L_{CDM}$ can improve the performance of the classic conditional flow matching, solidifying the benefits of our proposed approach.
>
> ---
>
> **Q4. How does FDM’s training time and memory footprint scale with data dimensionality, especially for high-resolution videos?**
>
> **Response:** Compared to the standard conditional flow matching (CFM), our proposed loss function involves an additional divergence term and essentially has a similar scalability as CFM. In practice, instead of explicitly computing the Jacobian matrix, we follow the approach used in [1] and [2], which employs the Hutchinson estimator to approximate the divergence term (see Lines 307-309 in our submission). Then, we only need to compute the derivative of a scalar-valued output with respect to the input to construct the loss, which involves a single call to torch.autograd.grad() and adds just one additional backward pass. Since we're only doing one extra backward pass, the additional memory footprint and computational time of computing the divergence loss scales similarly to that of the original CFM loss with respect to the data dimensionality. We compared the training times and peak memory usage, and found that incorporating the divergence loss does not significantly hinder the model's usability, with computational time and memory footprint remaining mostly within 1.5 times that of the original CFM.
>
> Regarding the video experiment mentioned by the reviewer, we first use a pretrained model to map the video data into a latent space. As a result, the computational cost of the flow matching model depends only on the dimensionality of the latent variables, not the raw video data.
>
> [1] Lu, Cheng, et al. "Maximum likelihood training for score-based diffusion odes by high order denoising score matching." International Conference on Machine Learning. ICML, 2022. https://arxiv.org/pdf/2206.08265
> [2] Lai, Chieh-Hsin, et al. "Fp-diffusion: Improving score-based diffusion models by enforcing the underlying score fokker-planck equation." International Conference on Machine Learning. ICML, 2023. https://arxiv.org/pdf/2210.04296
>
> ---
>
> **Q5. Can FDM be applied to non-Gaussian or non-OT probability paths?**
>
> **Response:** Yes, FDM can be applied to non-Gaussian paths. In our DNA sequence generation experiment, we use a Dirichlet probability path, which was designed in [3].
>
> [3] Stark et al. Dirichlet Flow Matching with Applications to DNA Sequence Design, ICML, 2024.
>
> ------
>
> Thank you for considering our rebuttal.

---

### Official Review · Reviewer_247t · 2025-03-17

**Overall Recommendation:** 3

**Summary:**

The paper proposes a modification to the flow matching / stochastic interpolant loss so as to better control the total variation distance between the model and the target at the final time of sampling, motivated by the fact that the standard loss is not sufficient to control the KL divergence (based on some assumptions on the target).

The modification they propose is a sort of "divergence matching" loss whereby the divergence of the model vector field is trained to match the divergence of the ground truth vector field defined by the interpolant.

They test the method on synthetic datasets used in previous works, trajectory sampling, dna sequencing, and video forecasting.

**Claims And Evidence:**

The motivation for the project is clear. They seek to control a divergence metric for their loss for a deterministic, ODE-based flow model. Other work has only done this using higher-order derivatives than the ones they consider here. However, the related work can get control on the KL, while this work is only on the TV. The authors acknowledge this fact.

One issue with the application of the method in the experiments that I'd like the authors to clarify: At the end of page 6, the authors state "we need to express the score function $\nabla \log p_t(x)$ learned by diffusion models in terms of the learned vector field $v_t (x, θ)$. This limits us to choosing a conditional probability path corresponding to an SDE with a known drift term $f$ and noise coefficient $g$." Note that this relation between the velocity field and the score *is only true if you have actually found the velocity field associated to the interpolant/probability path you specified*. If you have not learned this, then the equation relating $v_t$ to $s_t$ is not valid. *Moreover*, this score only corresponds to the actual $\nabla \log p_t$ if the velocity is exactly learned. So it is not clear that the equation at the bottom of page 6 is an exact relation, and would introduce extra errors.

**Essential References Not Discussed:**

The authors should probably also cite Liu et al, Flow Straight and Fast (2023) for its contributions to the flow matching literature as well.

**Experimental Designs Or Analyses:**

There is a wide breadth of experiments based on existing applications of generative models that people have tried.  The method seems to preform better, but only marginally. A question, then, is: given the fact that loss is more expensive (i.e you have to compute and match divergences), how does one measure the advantage of the technique properly? Can

In addition, how does the method compare for forecasting videos to: https://arxiv.org/pdf/2403.13724 ? It seems they report much lower FIDs, but also claim that Latent FM gets lower FIDs as well. It is probably worth citing this relevant literature, which seems to excel at this task.

**Methods And Evaluation Criteria:**

The proposed benchmarks make sense. They are basically doing a set of controlled trials comparing the standard method to their modification. It would also be nice to see how this performs versus the broader literature, though.

**Other Comments Or Suggestions:**

n/a

**Other Strengths And Weaknesses:**

In general, it would be nice to have a better understanding of the memory footprint given the divergence you have to compute. Does the method work when replaced with a hutchinson estimator, or is it too noisy?

**Questions For Authors:**

n/a

**Relation To Broader Scientific Literature:**

The contribution is to describe an avenue for improving a very commonly used generative modeling paradigm. There is not existing work as far as I know on divergence matching this way.

**Theoretical Claims:**

I checked the correctness of the proofs. They seem like decently straightforward applications of Jensen and Young's inequality.

One thing I was looking for in particular but did not necessarily find: The bounds rely on an analytic form for the score $\nabla \log p_t(x)$ arising from solving the continuity equation up to time $t$. This quantity is *not* equivalent to the score arising from the model i.e. $\hat s_t \neq \nabla \log p_t$, nor is it something that can be written down by relating a learned velocity field to a score.

---

> ### Author Rebuttal · Authors · 2025-03-30
>
> Thank you for your thoughtful review and valuable feedback. We have revised the paper according to all reviewers’ feedback. In what follows, we provide point-by-point responses to your comments.
>
> ---
>
> **Q1. At the end of page 6, … it is not clear that the equation at the bottom of page 6 is an exact relation, and would introduce extra errors.**
>
> **Response:** Thank you for your comment, which helped us identify areas where we can improve our presentation, and we have revised the paper accordingly.
>
> To address your concern, we stress that our theoretical results do not depend on this relation for the learned vector field $\mathbf{v}_t(x,\theta)$. As the reviewer noted, this relation holds only for the ground-truth vector field $\mathbf{u}\_t$ and the score function of its corresponding probability flow $p_t$ (due to the uniqueness of solutions to the continuity equation). Our derivations rely solely on the continuity equation and initial conditions, not on this specific relationship, which aligns with the reviewer’s observation.
>
> Our original intent in introducing this relation was to define a flow-matching model with the vector field $\mathbf{u}\_t$ corresponding to the score-based diffusion model with drift term $\bf f$ and noise coefficient $g$. Notice that approximating $\mathbf{u}\_t$ with a parameterized vector field $\mathbf{v}\_t(x,\theta)$ is enough to generate data effectively. However, for the numerical results (e.g., conditional probability computations in Table 2), we require an approximation of the score function $\nabla \log p_t$. Instead of computing $\nabla \log \hat p\_t$, we substitute the learned vector field $\mathbf{v}\_t(x,\theta)$ into the following relation to estimate the score:
> $$\nabla \log p\_{1-t}(x) = 2\frac{\mathbf{u}\_t(x)+{\bf f}\_{1-t}(x) }{g^2_{1-t}}.$$
> While this introduces an approximation error, it is a practical trade-off for computational feasibility.
>
> ---
>
> **Q2. The bounds rely on an analytic form for the score $\nabla\log p_t(x)$ arising from solving the continuity equation up to time $t$. This quantity is not equivalent to the score arising from the model i.e. $\hat s_t\neq \nabla \log p_t$, nor is it something that can be written down by relating a learned velocity field to a score.**
>
> **Response:** Indeed, computing $L_{DM}$ requires $\nabla\log p_t(x)$, which is derived by solving the continuity equation. To address this challenge, we introduce its conditional variant $L_{CDM}$. For $L_{CDM}$, we only need access to $\nabla\log p_t(x|x_1)$. In practice, we design $p_t(x|x_1)$ such that this quantity is readily available by plugging in values, bypassing solving an ODE explicitly. This approach ensures computational feasibility while maintaining the integrity of our bounds; see Theorem 4.2.
>
> ---
>
> **Q3. The method seems to perform better, but only marginally. Given the fact that loss is more expensive …, how does one measure the advantage of the technique properly?**
>
> **Response:** The improvements for all tasks except trajectory sampling for dynamical systems are way over twice the standard deviation. For dynamical systems, we further perform a t-test with the null hypothesis that the mean negative log-likelihoods (NLLs) computed by two distinct models are equal. We use 32,000 test trajectories to compute each mean NLL. The t-test confirms that the improvement in the likelihood estimation of FDM over FM is significant.
>
> Our approach introduces additional computational cost, but it gives better generation results and better likelihood estimation. For tasks requiring accurate likelihood estimation, our approach is especially more valuable compared to the baseline flow matching.
>
> ---
>
> **Q4. How does the method compare for forecasting videos to [1]?**
>
> **Response:** Thank you. We have cited this paper in the revision.
>
> Indeed, the results in [1] are impressive. We only found the codes for Navier-Stokes and CIFAR at github.com/interpolants/forecasting, which is the reason we did not include this approach as a benchmark in our study.
>
> [1] https://arxiv.org/pdf/2403.13724
>
> ---
>
> **Q5. The authors should probably also cite Liu et al, Flow Straight and Fast (2023) for its contributions to the flow matching literature as well.**
>
> **Response:** We have cited this paper in the revision.
>
> ---
>
> **Q6. Memory footprint in computing divergence. Does the method work when replaced with a Hutchinson estimator?**
>
> **Response:** In our implementation, we use the Hutchinson estimator to approximate the divergence term; see Lines 307-309 in our submission. It works well and does not significantly raise the memory footprint, which consistently stays within 1.5×.
>
> For the 2D tasks, we test both computing the Hutchinson divergence estimator and the exact one; both result in nearly identical results. We use the Hutchinson estimator in all the other experiments due to memory constraints, following [2].
>
> [2] https://arxiv.org/pdf/2206.08265
>
> ---
>
> Thank you for considering our rebuttal.

---

> > ### Comment · Reviewer_247t · 2025-04-04
> >
> > Thanks for the information. I'll maintain my score, as it is already a weak accept!

---

### Official Review · Reviewer_uLYh · 2025-03-18

**Overall Recommendation:** 3

**Summary:**

The paper seeks to use PDEs to construct a theoretical bound on flow matching, and improve upon it using said insight by adding a divergence mismatch to the loss term which improves upon the probability path. They construct experiments on simple generative examples, along with DNA sequence generation and video prediction and show that it improves upon existing methods.

**Claims And Evidence:**

Their claims seem to be sufficiently supported mathematically and experimentally. I would like to see the performance of their method on some more diverse video datasets or standard image benchmarks for generative modeling (CIFAR, Imagenet).

**Essential References Not Discussed:**

None.

**Experimental Designs Or Analyses:**

Specific experiment soundness seem valid and include error bars for some of the experiments, which seems sufficient.

**Methods And Evaluation Criteria:**

The non-toy experiments seem to be the DNA sequence generation and the KTH dataset. The KTH dataset isn't used usually for generative modeling benchmarks due to its limited diversity in terms of samples. It works as a simple example but some more diverse motion datasets such as BAIR (as done in Davtyan et. al (2023)) could be more interesting. Evaluation criteria seem to be the de-facto norm for all the experiments.

**Other Comments Or Suggestions:**

Minor typos (does not effect valuation)
L_{FDM} is used in the graph of Figure 1, but caption reads L_{DM}

**Other Strengths And Weaknesses:**

None.

**Questions For Authors:**

None.

**Relation To Broader Scientific Literature:**

The paper offers some key insights for flow-based generative modeling by refining the flow matching approach, which also offer enhanced likelihood estimation. They also demonstrate their strengths with applications in structured data such as dynamical systems, dna sequences, and videos.

**Theoretical Claims:**

I checked section 3 for correctness, and the claims seem to be valid.

---

> ### Author Rebuttal · Authors · 2025-03-30
>
> Thank you for your thoughtful review and valuable feedback. We have revised the paper according to all reviewers’ feedback. In what follows, we provide point-by-point responses to your comments.
>
> ----
>
> **Q1. I would like to see the performance of their method on some more diverse video datasets or standard image benchmarks for generative modeling (CIFAR, Imagenet).**
>
> **Response:** During the rebuttal period, we conducted experiments to showcase the efficacy of our proposed FDM in improving flow matching (FM) for these reviewer-mentioned tasks. We first compare the performance of FDM against the baseline conditional FM (CFM) for generative modeling of CIFAR10 here, and we will show the advantages of FDM for other video generation tasks in **Q2**.
>
> We follow the experimental settings in Appendix E of the flow matching baseline paper [1] to train the model for CIFAR10 generation. Due to the time constraint, we only compare our FDM against CFM for CIFAR10 generation using the better-performing optimal transport (OT) path in the baseline paper [1]. The following results confirm the advantages of FDM.
>
> |  Method  |  NLL ↓   | FID ↓   |
> |-------------|------------|-----------|
> | CFM       |   2.99    |   6.35   |
> | **FDM (ours)**       |   **2.85**    | **5.62**    |
>
> [1] https://arxiv.org/pdf/2210.02747
>
> ---
>
> **Q2. The KTH dataset isn't used usually for generative modeling benchmarks due to its limited diversity in terms of samples. It works as a simple example but some more diverse motion datasets such as BAIR (as done in Davtyan et. al (2023)) could be more interesting.**
>
> **Response:** We appreciate your feedback. For the BAIR dataset, we predict 15 future frames based on a single initial frame, with each frame having a resolution of $64 \times 64$ pixels. Due to the highly stochastic motion in the videos of the BAIR dataset, we evaluate the model as follows (following [2]): We randomly select 256 test videos, and generate 100 samples per test video where each model is conditioned on the same initial frame of the video. Finally, we compute the FVD for each model comparing the $256\times 100$ generated samples against the 256 test videos. Due to time constraints, we omit training the frame refinement network, which operates independently of the main model but could potentially enhance sample quality. The result is as follows:
>
> | Method                  | FVD ↓           | Memory (GB) | Time (hours) |
> |-------------------------|------------------|-----------|----------------|
> | TriVD-GAN-FP            | 103              | 1024      | 280            |
> | Video Transformer       | 94               | 512       | 336            |
> | LVT                     | 126              | 128       | 48             |
> | RaMViD (Diffusion)      | 84               | 320       | 72             |
> | Latent FM           | 146              | 24.2      | 25             |
> | **Latent FDM (ours)**   | **123 ± 4.5**    | 35        | 36             |
>
> As mentioned in [2], many models for the BAIR task are computationally expensive, whereas latent FM achieves a favorable trade-off between FVD and computational cost. Our approach further improves latent FM with acceptable additional computational overhead.
>
> [2] https://arxiv.org/abs/2211.14575
>
> ----
>
> **Q3. Minor typos (does not effect valuation) L_{FDM} is used in the graph of Figure 1, but caption reads L_{DM}**
>
> **Response:** Thank you. We have fixed this in the revised manuscript.
>
>
> ------
>
> Thank you for considering our rebuttal.

---

### Decision · Program_Chairs · 2025-05-01

**Decision:**

Accept (poster)

**Comment:**

The proposed method is well motivated and it is theoretically supported. The empirical evaluation indicates limited improvements over the baselines.